# Factor H binding proteins protect division septa on encapsulated *Streptococcus pneumoniae* against complement C3b deposition and amplification

Anuj Pathak[1], Jan Bergstrand[2], Vicky Sender[1], Laura Spelmink[1], Marie-Stephanie Aschtgen[1], Sandra Muschiol[1], Jerker Widengren[2] & Birgitta Henriques-Normark[1,3,4]

*Streptococcus pneumoniae* evades C3-mediated opsonization and effector functions by expressing an immuno-protective polysaccharide capsule and Factor H (FH)-binding proteins. Here we use super-resolution microscopy, mutants and functional analysis to show how these two defense mechanisms are functionally and spatially coordinated on the bacterial cell surface. We show that the pneumococcal capsule is less abundant at the cell wall septum, providing C3/C3b entry to underlying nucleophilic targets. Evasion of C3b deposition at division septa and lateral amplification underneath the capsule requires localization of the FH-binding protein PspC at division sites. Most pneumococcal strains have one PspC protein, but successful lineages in colonization and disease may have two, PspC1 and PspC2, that we show affect virulence differently. We find that spatial localization of these FH-recruiting proteins relative to division septa and capsular layer is instrumental for pneumococci to resist complement-mediated opsonophagocytosis, formation of membrane-attack complexes, and for the function as adhesins.

---

[1] Department of Microbiology, Tumor and Cell Biology, Karolinska Institutet, SE-171 77 Stockholm, Sweden. [2] Department Applied Physics, Royal Institute of Technology (KTH), Experimental Biomolecular Physics, SE-106 91 Stockholm, Sweden. [3] Lee Kong Chian School of Medicine (LKC) and Singapore Centre on Environmental Life Sciences Engineering (SCELSE), Nanyang Technological University, Singapore 639798, Singapore. [4] Department of Clinical Microbiology, Karolinska University Hospital, SE-171 76 Stockholm, Sweden. Correspondence and requests for materials should be addressed to B.H-N. (email: birgitta.henriques@ki.se)

The invasive respiratory pathogens, *Haemophilus influenzae*, *Neisseria meningitidis*, and *Streptococcus pneumoniae*, have evolved similar strategies, such as the expression of polysaccharide capsules and complement Factor H (FH) recruiting surface proteins, to evade deposition of C3b and activation of the alternative pathway of the complement system[1–3]. The pneumococcal polysaccharide capsule is critical for virulence and is known to inhibit complement activity[4]. Localization sites of deposited complement on the bacteria in relation to the capsule are not known. The capsule is believed to form an external shield for C3 entry and for the covalent binding of C3b to underlying nucleophilic targets[5–7]. So far more than 97 capsular polysaccharides have been described, and all except for serotypes 3 and 37, are covalently anchored to the bacterial peptidoglycan via the wzy pathway[6]. The ovoid shape of pneumococci results from a combination of lateral and septal peptidoglycan synthesis. Pneumococcal growth occurs by formation of the lateral wall via a complex protein machinery referred to as the elongasome, while the septal wall is formed by a second machinery, which is assembled at the position for the FtsZ-ring[8], the divisome[9]. The divisome drives membrane invagination and septal wall formation. To generate a new bacterial pole the septal wall needs to be split, requiring cell wall hydrolytic enzymes[10], resulting in the formation of two new poles[11]. This process is highly variable leading to pneumococcal populations consisting of single cocci, diplococci and chains of different lengths[12].

The pneumococcal peptidoglycan not only contains covalently anchored capsular polysaccharide, but also a set of surface proteins carrying a LPxTG motif allowing sortase-mediated covalent linkage to lysine residues on the peptidoglycan stem peptides[13]. Another set of pneumococcal surface proteins, the choline-binding proteins, contain choline-binding motifs recognizing choline residues on teichoic acids of which wall teichoic acids (WTA) are covalently anchored to the peptidoglycan, while lipoteichoic acids (LTA) are anchored in the cytoplasmic membrane[14]. It is not known if there are differences in level and composition of cell wall associated macromolecules between the lateral and septal wall in the pneumococcus.

The pneumococcal choline-binding protein PspC (CbpA) recruits FH from human serum that selectively inhibits amplification of deposited C3b, and accelerates the decay of the alternative pathway C3 convertase[15–17]. Apart from FH recruitment, different allelic forms of PspC have also been reported to interact with host factors like pIgR, vitronectin and SIgA[18–20]. While most pneumococcal strains carry one gene of *pspC*, the pneumococcal lineage CC138, successful in causing colonization and invasive disease, contains two closely linked *pspC* genes, *pspC1* encodes a conventional choline-binding PspC (denoted PspC1 in CC138), and *pspC2* a cell wall anchored LPxTG version of PspC (denoted PspC2), both able to bind human FH[21–23]. PspC2 lacks the motif responsible for pIgR interaction[24].

In the present study, we combine super-resolution imaging techniques, mutants affecting protein localization, and functional analyses to show that the division septum represents the pneumococcal Achilles heel in its capsular barrier defense against complement C3b deposition. To cope with the low content of capsular polysaccharide at division septa, pneumococci have evolved FH binding proteins localized at division sites, and allow complement entry while division septa are formed at these sites. We show that the spatial positioning of virulence-associated cell wall proteins such as PspC, relative to the division septum and the capsular layer, have profound implications for defence against complement-mediated opsonophagocytosis, and formation of membrane attack complexes (MACs), needed for bacteria in an inflamed environment. Our data also demonstrate that a complement evasive protein, depending on accessibility outside the capsular layer, may mediate bacterial attachment to epithelial cells, potentially favouring healthy colonization.

## Results

**C3b deposition occurs at or close to division septa.** Encapsulated pneumococcal strains of serotypes 4 (TIGR4), 2 (D39), and 6B (BHN418) were incubated with human serum, and deposited complement C3b was monitored by anti-C3b staining (Supplementary Table 1, Fig. 1a). Using confocal microscopy, C3b antibodies were in each of the three strains shown to recognize deposited C3b as discrete bands on the cells (Fig. 1a). TEM images performed on the serotype 6B strain BHN418 revealed bulky complement deposits precisely localized at some but not all division septa (18 out of 80 visible septa), the latter seen as thin electron-dense bands (Fig. 1b). Deposition of C3b was also confirmed by immunogold staining and TEM (Supplementary Fig 1). When the isogenic non-encapsulated mutant BHN418Δcapsule was examined by TEM, C3b was found to be deposited all around the cells, and immunostaining with C3b antibodies revealed the same uniform staining pattern (Fig.1c, d). SEM images of encapsulated BHN418 showed regularly spaced elevations on the bacteria that were absent at some division septa (Fig. 1e, arrows). As these elevations were completely absent in the capsular mutant BHN418Δcapsule (Fig. 1f) we suggest that they represent the cell wall associated serotype 6B capsule that appears less abundant at division septa.

**C3b and C5b-9 localize at division septa underneath the capsule.** To investigate localization patterns of the capsule and C3b in strains TIGR4, D39, and BHN418 we used super-resolution stimulated emission depletion (STED) microscopy and performed double staining. We observed that C3b deposition occurred mainly as distinct bands (rings) at division septa, possibly due to less capsule in these areas as observed using SEM microscopy (Figs. 2a, 1e). The edges of these C3b bands were localized underneath the capsular layer in all three strains (Fig. 2a). C3b deposition on bacteria can result in formation of MAC[25,26]. We therefore investigated MAC complex formation in BHN418 using a specific antibody, detecting a neo-epitope not present in monomeric C9 and only present in polymeric C9. Immunofluorescence microscopy of C5b-9 (MAC) staining of the bacteria revealed a band pattern in a few bacteria at or close to division septa (Fig. 2b). While 15% (19 out of 124 cells counted) of BHN418 cells showed a C3b signal, only 5% (8 out of 173 counted) harbored C5b-9. This suggests that formation of MAC complexes does not occur at each division septum where C3b is deposited. C5b-9 (MAC) is the terminal product of the complement cascade, which requires longer stable complement deposition. Probably blockade at most septa must be so efficient that the cascade does not always progress. By comparing STED images of C3b and C5b-9 deposition there was a larger distance between the capsule and the edges of C5b-9 bands than between the capsule and C3b (Fig. 2c, d, Supplementary Fig 2). This suggests that the major portion of C3b in wild type pneumococci becomes deposited in or around the septal wall, whereas the C5b-9 MAC complexes are formed in the membrane beneath the wall.

**Factor H recruitment occurs at distinct sites underneath the capsular layer.** Pneumococci are known to prevent C3b deposition and amplification by recruitment of Factor H (FH). Exposing encapsulated strains, TIGR4, D39, and BHN418, to human FH and performing double staining with the respective capsular

antibody and human FH-antibodies, revealed that FH was deposited at distinct sites in virtually all cells (Fig. 3a). Furthermore, examining BHN418 cells with STED microscopy suggested that FH was mainly recruited to sites localized underneath the capsular layer (Fig. 3b).

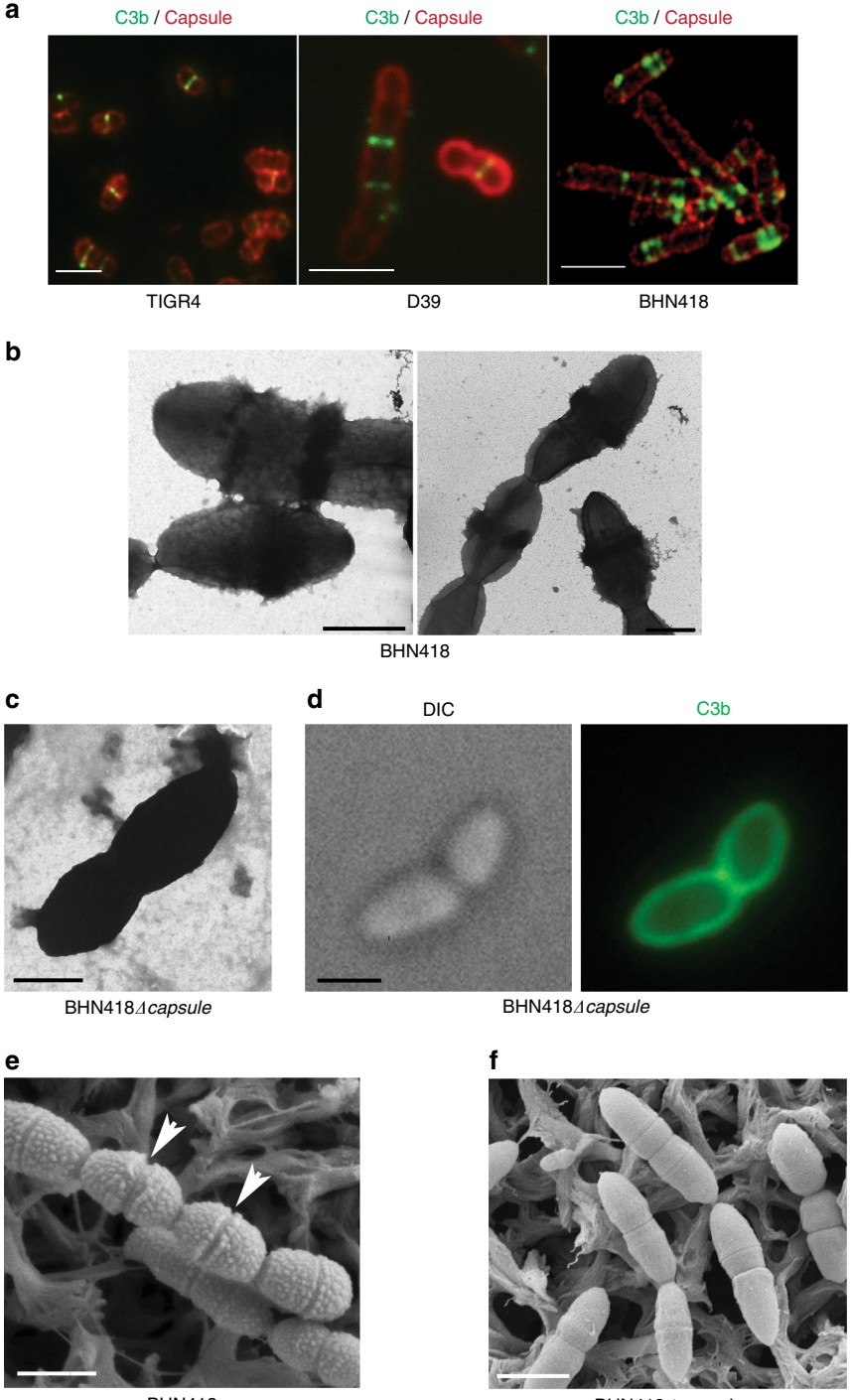

**Fig. 1** Complement C3b deposition occurs at or close to division septa in encapsulated *Streptococcus pneumoniae*. **a** Representative immunofluorescence images showing distinct septal localization of C3b on pneumococcal strains TIGR4 (serotype 4), D39 (serotype 2) and BHN418 (serotype 6B). C3b was stained using goat anti-C3 antibody followed by incubation with anti-goat Alexa fluor 488 secondary antibody (green). The capsule was detected using respective rabbit anti-capsule serum and anti-rabbit Alexa fluor 594 antibody (red). Scale bar = 5 μm. **b** Representative TEM images of wt BHN418 after incubation with 20% normal human serum. C3b deposits are seen as dark rings at division septa. **c** Representative TEM image of BHN418Δ*capsule* after incubation with 20% normal human serum. A uniform deposit of C3b is observed. **d** Representative immunofluorescence images of C3b deposition on BHN418Δ*capsule* after incubation with 20% normal human serum. C3b was stained using goat anti-C3 antibody followed by incubation with anti-goat Alexa fluor 488 secondary antibody (green). **e** SEM images of wt BHN418. Arrows indicate two division septa lacking surface humps representing the capsule. **f** SEM images of BHN418Δ*capsule* devoid of surface humps. **b**–**f** Scale bar = 1 μm

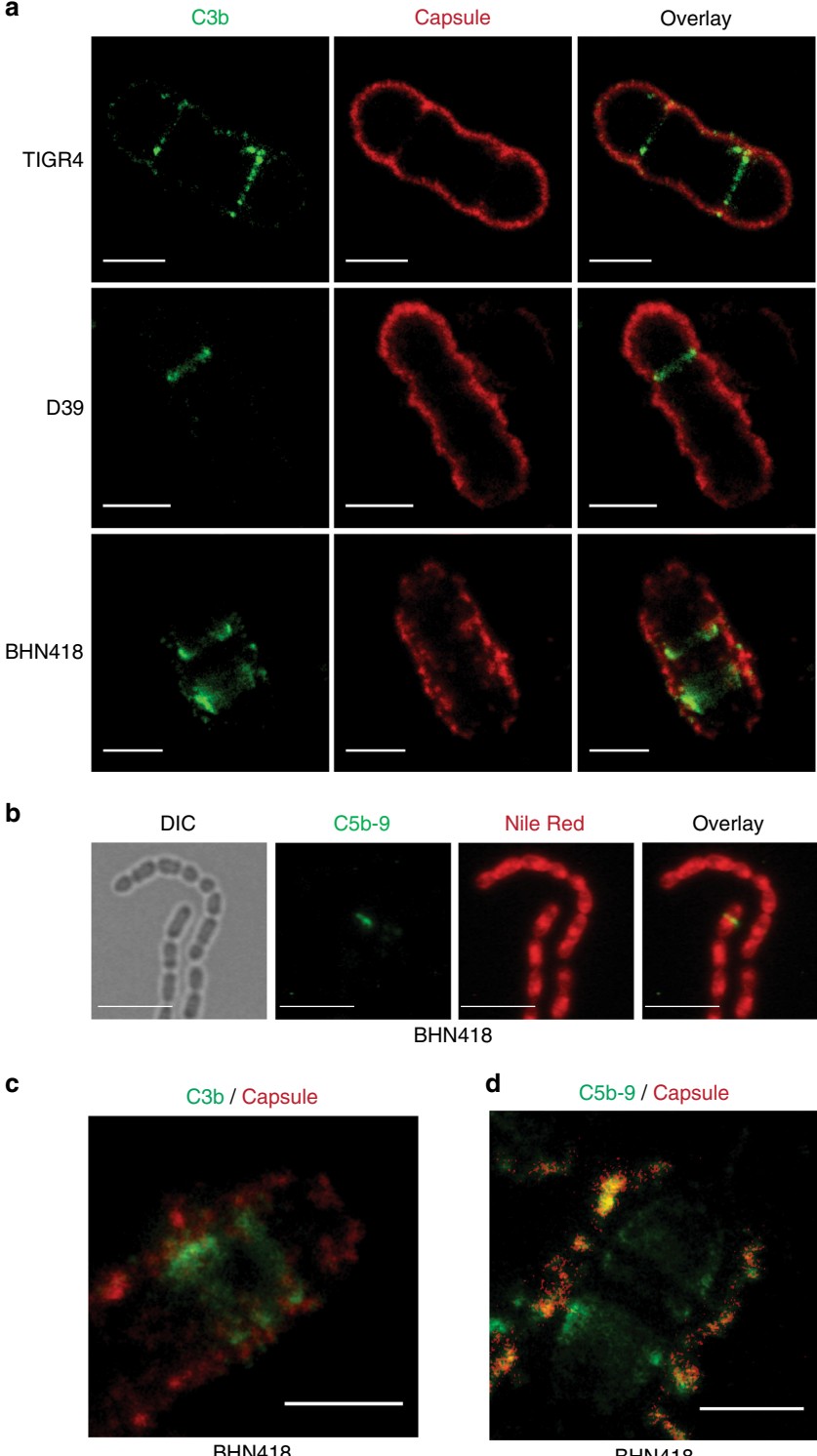

**Fig. 2** C3b and C5b-9 MAC complexes localize underneath the capsular layer. **a** Super-resolution STED microscopy images of strains TIGR4, D39, and BHN418 after incubation with 20% normal human serum. Bacteria were sequentially stained with goat anti-C3 antibody followed by Atto647N labeled secondary antibody (green). The capsule was detected using respective rabbit anti-capsule serum and anti-rabbit Alexa fluor 594 antibody (red). Scale bar = 1 μm. **b** Representative immunofluorescence images of C5b-9 deposition on strain BHN418 after incubation with 20% normal human serum. Bacteria were sequentially stained with mouse anti-C5b-9 antibody and anti-mouse Alexa fluor 488 labeled secondary antibody (green). Bacterial membranes were stained with nile red (red). Scale bar = 5 μm. **c** Representative super-resolution STED microscopy image of C3b deposition on strain BHN418 after incubation with 20% normal human serum. Bacteria were stained for C3b and the capsule as in **a**. **d** Representative super-resolution STED microscopy image of C5b-9 deposition on strain BHN418 after incubation with 20% normal human serum. Bacteria were sequentially stained with mouse anti-C5b-9 antibody followed by Atto647N labeled secondary antibody (green). The capsule was detected using rabbit anti-6B capsule serum and anti-rabbit Alexa fluor 594 antibody (red). **c, d** Scale bar = 1 μm

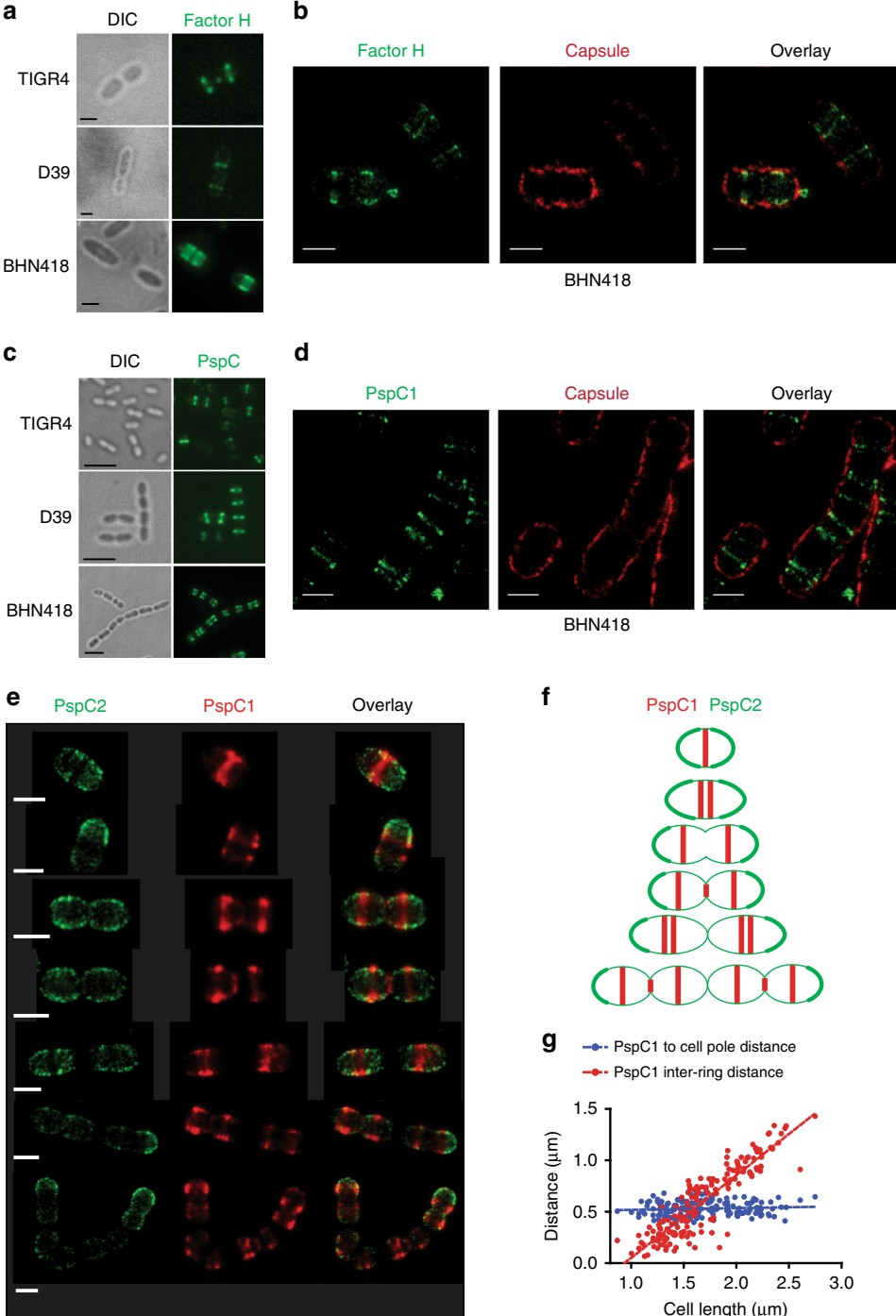

**Fig. 3** Factor H (FH) recruitment by encapsulated pneumococci occurs at division sites where FH binding proteins are localized. PspC1 and PspC2 localize differently on the pneumococcal cell. **a** Representative immunofluorescence images of distinct localization of FH on strains TIGR4, D39, and BHN418. After incubation with pure FH, bacteria were stained with goat anti-FH antibody and FITC-labeled secondary antibody (green). **b** Super-resolution STED microscopy images of FH (stained green) and capsule (stained red) on strain BHN418. **a**, **b** Scale bar = 1 μm. **c** Representative immunofluorescence images of PspC localization on strains TIGR4, D39, and BHN418. PspC was detected with mouse anti-PspC antiserum (D39) followed by anti-mouse Alexa fluor 488 (green) secondary antibody. Scale bar = 5 μm. **d** Super-resolution STED microscopy images of PspC1 (green) and the capsule (red) on strain BHN418. **e** Super-resolution STED microscopy images of different growth phases of strain BHN418 showing the distribution patterns for PspC1 (red) and PspC2 (green) during cell division and cell growth. **d**–**f** Scale bar = 1 μm. **f** Schematic representation of the PspC1 and PspC2 distribution during different stages of cell division and cell growth. **g** The distances between PspC1 rings, and between PspC1 rings and the closest pole in BHN418 (n = 166) are shown. Linear fitting curve is shown in the plot

Most clinical isolates of *Streptococcus pneumoniae* express a FH recruiting protein PspC (CbpA) that is anchored to choline residues on either LTA or WTA[3]. Immunofluorescence images using PspC antibodies revealed that PspC is positioned as one distinct band per single coccus in a chain (Fig. 3c) in all the three pneumococcal strains examined (TIGR4, D39, and BHN418). For the serotype 6B strain BHN418, we further found by STED microscopy that its choline-binding PspC1 protein preferentially interacts with its antibody underneath the capsular layer (Fig. 3d). As a control for protein localization relative to the capsule layer we made use of antibodies to RrgB, the major protein of adhesive pilus-1 polymers, and demonstrate that this LPxTG protein can be recognized both outside, within and beneath the capsular layer (Supplementary Fig 3).

**PspC1 localizes at division sites and PspC2 mainly at the poles**. We studied the interaction between FH and the two PspCs (PspC1 and PspC2) found in the 6B strain BHN418, belonging to the successfully spread clonal lineage CC138[22] (Supplementary Fig 4 and Supplementary Data 1). We found that purified recombinant PspC1 and PspC2 both bound human FH with similar binding strengths, as determined by surface plasmon resonance/BIACORE analysis (Supplementary Table 2, Supplementary Fig 5). We also used peptide mapping to localize the interaction regions between FH and the two PspC proteins (Supplementary Fig 6a). We further performed sequence alignments of the FH binding domains of different PspC proteins (Supplementary Fig 6b) and showed that PspC1 and PspC2 belong to two different families of PspCs, with PspC2 belonging to the same family A as PspC in TIGR4, previously crystalized in the presence of FH[27], and PspC1 to family B including strain D39, which has been used in several FH binding studies[28].

We next determined the localization of PspC1 and PspC2 with super-resolution STED microscopy in exponentially growing BHN418 bacteria, using polyclonal antibodies raised against PspC1 (PspC from D39) and PspC2 (from BHN418), that were specific to their respective protein (Supplementary Fig 7). In newborn single cocci, PspC1 localized in the form of a dense band (ring) about 200 nm in width (180 nm ± 50 nm, $n = 97$), which tended to split into two that moved apart during elongation of the bacterium. A third thin PspC1 band could also be observed at the septum of dividing diplococci (Fig. 3e, f). The distance between a PspC1 band and its older pole was the same, about 0.5 μm (0.50 ± 0.02), irrespective of cell length, suggesting that all cell wall elongation occurs between the PspC1 rings, and that PspC1 localizes at the division site, marking where the next division septum will be formed (Fig. 3g). PspC2 was not found at the division sites, but localized preferentially at the bacterial poles (Fig. 3e). Newborn cocci were fully covered by the two proteins as PspC1 decorated the central area not covered by PspC2. In diplococci and chains, however, constriction sites appeared that were not decorated by either of the two proteins. Even though the two PspC proteins were differentially localized, double staining with PspC2 and serotype 6B antibodies suggested that also the major portion of PspC2 molecules localized underneath the capsular layer (Supplementary Fig 8).

**PspC1 is the major contributor to FH recruitment**. We used strain BHN418 to create deletion mutants generating BHN418ΔpspC1, BHN418ΔpspC2, and the double mutant BHN418ΔpspC1ΔpspC2. We first analyzed FH binding using flow cytometry (Fig. 4a). The BHN418 double mutant BHN418ΔpspC1ΔpspC2 showed no FH binding, demonstrating that these two PspC proteins are the sole contributors to FH binding in strain BHN418. Furthermore, absence of PspC1 had a

considerably larger negative effect on FH recruitment than the absence of PspC2 (Fig. 4a). Co-staining of FH and PspC1 on BHN418 showed that FH deposition preferentially occurs in a banding pattern similar to that of PspC1 (Fig. 4b), and not where PspC2 is localized (Fig. 4c). However, co-staining of FH and PspC2 in BHN418ΔpspC1 showed that the FH signal in this mutant fully co-localized with the PspC2 signal, suggesting that in vivo absence of PspC1 leads to that PspC2 becomes fully available for FH recruitment (Fig. 4d). PspC1 is a choline-binding protein and can therefore be removed by choline chloride treatment, unlike PspC2 that is covalently linked to the cell wall by its LPxTG motif. Choline chloride treatment of BHN418 and its *pspC* mutants as well as of another encapsulated CC138 isolate (BHN191) made FH in serum available for recruitment by polar located PspC2 (Fig. 4e, Supplementary Fig 9). Absence of PspC2 in BHN418ΔpspC2 had no effect on the pattern of FH deposition that like the wild type remained fully co-localized with PspC1 (Fig. 4f). All strains used in the study showed similar FH binding pattern when 20% normal human serum (NHS) was used as a serum source instead of pure FH (Supplementary Fig 10). Images were also analyzed by calculating the co-localization constant between PspC1 and PspC2, and FH, showing low co-localization constants between FH and PspC2 in wild type bacteria, but a higher constant in bacteria lacking PspC1. Also, the co-localization constant was higher for PspC1 and FH in wt BHN418 bacteria (Supplementary Fig 11)

**PspC1 protects division sites from C3b deposition and MAC complex formation**. We next used STED microscopy and double staining of the capsule and C3b to visualize the location of the two FH-recruiting PspC proteins (PspC1 and PspC2) in relation to C3b deposition and amplification. Wild type BHN418 and its *pspC* mutant derivatives were incubated in human serum containing FH and C3 (Fig. 5a). In the absence of PspC1, considerably more cells (64%) contained deposited C3b (97 positive out of 150 counted) compared to wt BHN418 (16%) (19 positive out of 124 counted). Importantly, a much larger area underneath the capsular layer contained deposited C3b when compared to the wt (Supplementary Fig 12). However, lateral C3b amplification within the cell wall did not involve the bacterial poles. In the absence of both PspC1 and PspC2, the bacterial poles also showed C3b deposition, and in some cells the entire area underneath the capsule was covered with C3b. In the presence of PspC1, absence of PspC2 had only a marginal effect on lateral C3b amplification. Together, these data strongly suggest that the initial deposition of C3b in encapsulated pneumococci occurs at bacterial division septa from where C3b amplification can spread laterally within the cell wall, underneath the capsular layer, unless PspC1 prevents this activation step by recruiting FH that likely also enters via the division septa. These results were confirmed by flow cytometry where loss of PspC1 showed enhanced C3b deposition compared to wt BHN418, whereas loss of PspC2 had a considerably smaller impact on C3b deposition (Fig. 5b). Images were also analyzed by calculating the co-localization constant between the capsular signal and the complement signal, showing a low co-localization constant between the capsule and C3b for all the strains tested (Supplementary Fig 13).

We also compared the formation of MACs in wt BHN418 and in the *pspC* mutants. Immunofluorescence microscopy of C5b-9 (MAC) staining of the bacteria revealed a band pattern on wt and mutant bacteria (Fig. 5c). However, in the absence of PspC1 considerably more cells stained positive for C5b-9, 40% (85 positive out of 213 counted), compared to about 5% for the wt (8 positive out of 173 counted) and the *pspC*2 mutant (9 positive out of 174 counted). Absence of both *pspC1* and *pspC2* had no

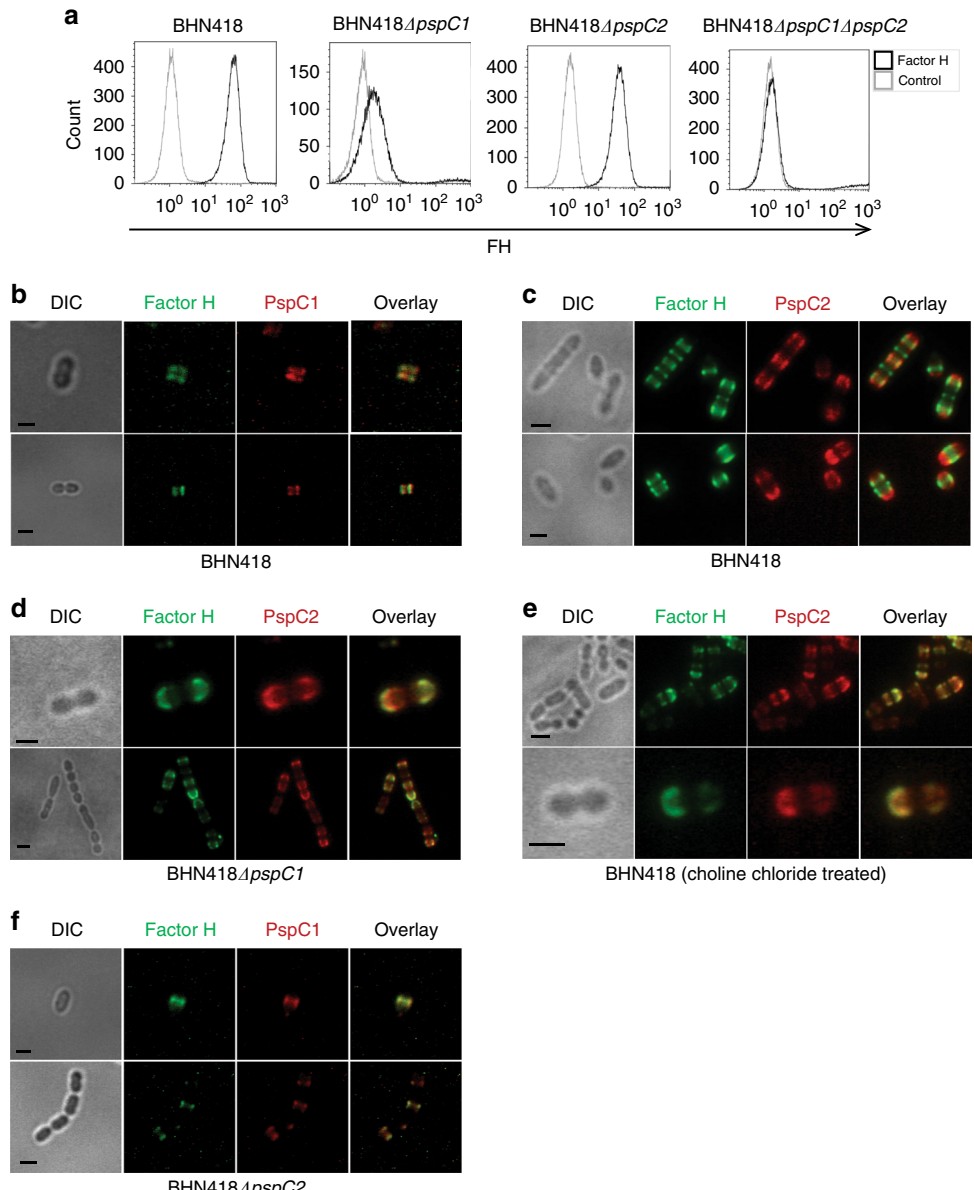

**Fig. 4** Even though PspC1 and PspC2 show similar binding strength to FH in vitro, PspC1 is the major contributor to FH recruitment in vivo. **a** Representative histogram of FH binding to BHN418 and its isogenic *pspC* deletion mutants using flow cytometry. Bacteria were incubated with purified human FH and stained as in Fig. 3a. Bacteria incubated without FH were used as a control for each strain. The histogram shown is representative of three independent experiments. **b** BHN418 was incubated with purified human FH, and FH was detected using a polyclonal goat anti-FH antibody and a FITC-labeled rabbit anti-goat IgG secondary antibody (green). Bacteria were then stained for PspC1 using anti-PspC1 antiserum and anti-mouse Alexa flour 594 secondary antibody (red). **c** Strain BHN418 was incubated with purified human FH, and FH was detected using a polyclonal goat anti-FH antibody and a FITC-labeled rabbit anti-goat IgG secondary antibody (green). Bacteria were then stained for PspC2 using serum purified rabbit anti-PspC2 polyclonal antibody labeled with Alexa fluor 594 (red). **d** The deletion mutant of PspC1, BHN418Δ*pspC1*, was stained for bound FH and PspC2 as in **c**. **e** Choline chloride treated BHN418 bacteria were incubated with purified human FH. FH and PspC2 were stained as in **c**. **f** The mutant strain BHN418Δ*pspC2* was stained for bound FH and PspC1 as in **b**. **b**–**f** Scale bar = 1 μm

additional effect on MAC-complex-formation. This observation was also confirmed by flow cytometry (Fig. 5d). Furthermore, we detected higher amounts of C3d deposited on wt BHN418 and on the *pspC2* mutant than on the *pspC1* mutant using western blot analysis, suggesting that degradation of complement components is more effective in the presence of PspC1 (Supplementary Fig 14, full blots are shown in Supplementary Fig 24). Together these data demonstrate that PspC1 at division sites effectively protects encapsulated *S. pneumoniae* against C3b deposition, and lateral C3b amplification in the cell wall, as well as prevents MAC complex formation in the cytoplasmic membrane.

**PspC1 at division sites protects encapsulated pneumococci against opsonophagocytosis.** The main effect of the complement system in encapsulated pneumococci is to enhance opsonopha-gocytosis mediated by deposited C3b interacting with the C3b-receptor on phagocytic cells[29]. We therefore followed bacterial uptake by the human-derived macrophage cell line THP-1 in the presence of human serum. Phagocytosis was considerably higher for BHN418 and its mutant derivatives when bacteria were pre-incubated with human serum as compared to untreated bacteria (Fig. 6a). Strain BHN418Δ*pspC1* was phagocytosed better than wt BHN418, which might be explained by the higher number of

division septa with C3b deposits, and the higher amounts of deposited C3b in the absence of PspC1. As expected, absence of PspC2 at the poles (BHN418ΔpspC2) had no effect on opsono-phagocytosis compared to the wt. This difference in phagocytosis between wt BHN418 and the *pspC1* mutant was not observed when bacteria were incubated with either pure FH or pure C3b (Supplementary Fig 15).

**PspC2 at the poles mediates adhesion to epithelial cells**. Surprisingly, the double mutant BHN418ΔpspC1ΔpspC2 was less phagocytosed as compared to the single mutant BHN418ΔpspC1 (Fig. 6a). This finding suggested that PspC2 might mediate

additional non-C3b dependent interactions with human cells. Pneumococcal CC138 strains of serotype 6B expressing PspC1 and PspC2 have demonstrated high efficiency to colonize the respiratory tract in children[22] suggesting high colonization ability. Moreover, pneumococcal PspC, besides playing a central role in FH recruitment, has also been shown to mediate adhesion to epithelial cells[30]. We therefore examined adhesion of strain BHN418 and its mutant *pspC* derivatives to human lung derived A549 cells using fluorescence microscopy. BHN418 was found to attach to A549 cells via the poles (Fig. 6b, c), whereas a random attachment pattern was found for the *pspC2* deletion mutant (Fig. 6c). Absence of PspC2, but not PspC1, resulted in a

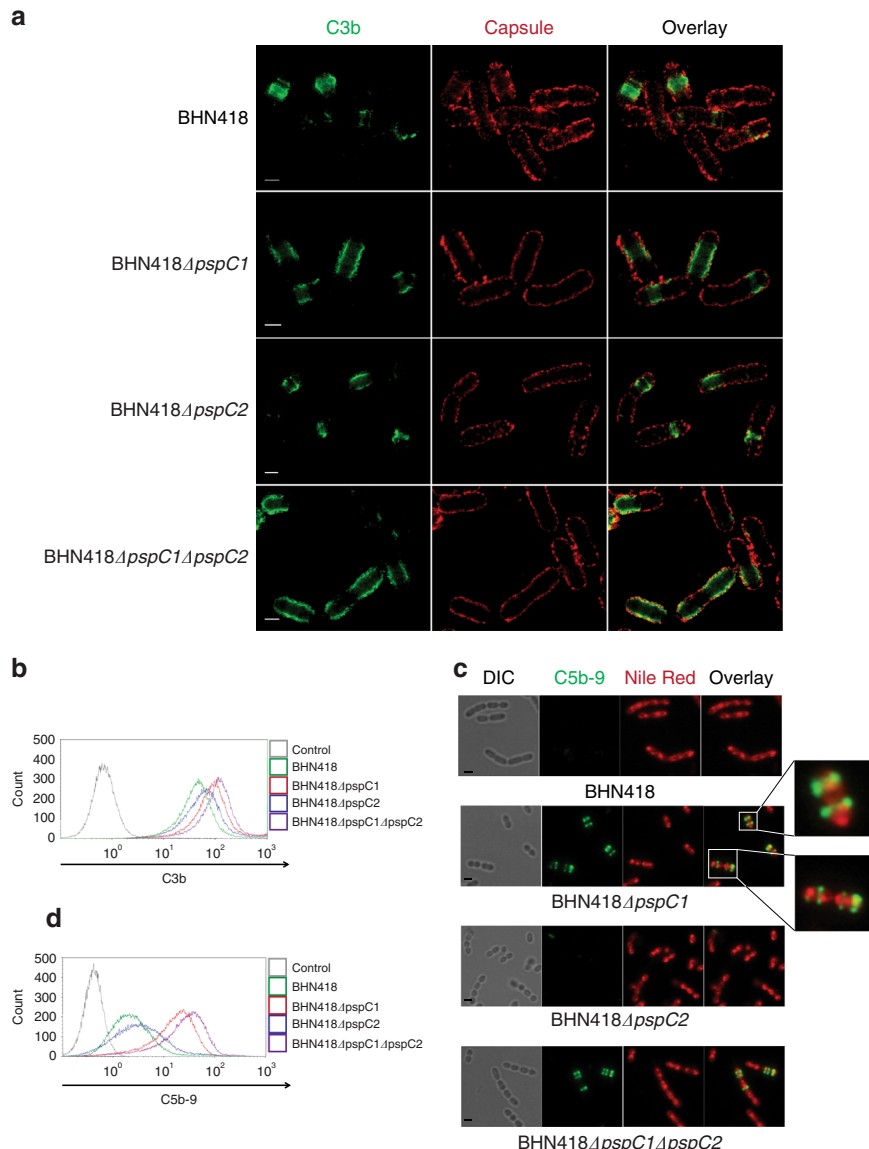

**Fig. 5** PspC1 protects division sites from C3b deposition and MAC complex formation, and prevents lateral C3b amplification within the cell wall. **a** Representative super-resolution STED microscopy images of C3b deposition on wt BHN418 and its isogenic *pspC* deletion mutants after incubation with 20% normal human serum. C3b and the capsule were detected as in Fig. 2a. Scale bar = 1 μm. **b** Representative histogram of C3b deposition on strain BHN418 and on its isogenic *pspC* deletion mutants after incubation with 20% normal human serum using flow cytometry. Bacteria were sequentially stained with goat anti-C3 antibody followed by FITC-labeled secondary antibody. **c** Representative immunofluorescence images of C5b-9 deposition on strain BHN418 and on its isogenic *pspC* deletion mutants after incubation with 20% normal human serum. Bacteria were stained as in Fig. 2b. Scale bar = 1 μm. **d** Representative histogram of C5b-9 deposition on strain BHN418 and on its isogenic *pspC* deletion mutants after incubation with 20% normal human serum using flow cytometry. Bacteria were sequentially stained with mouse anti-C5b-9 antibody followed by anti-mouse Alexa fluor 488 labeled secondary antibody

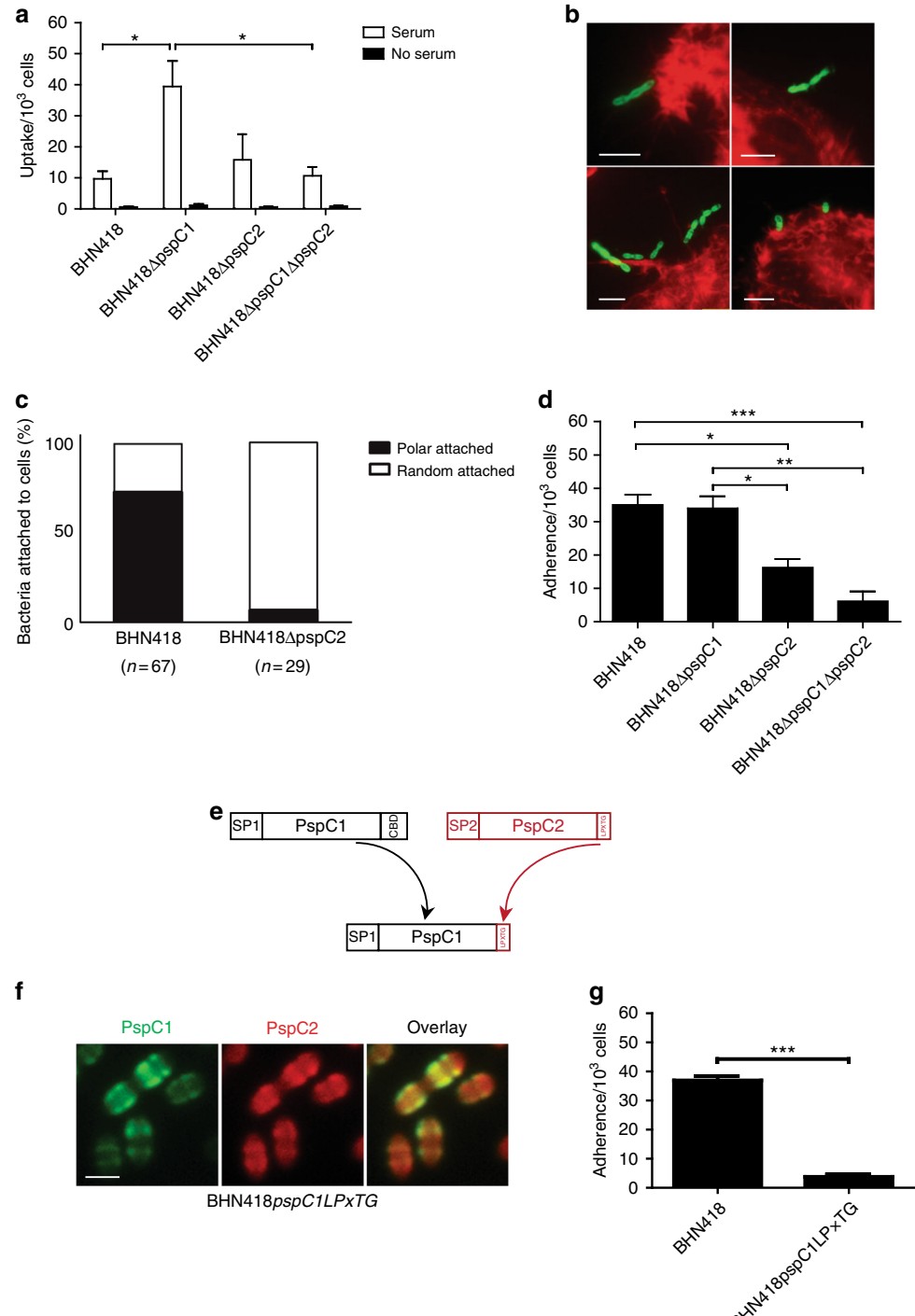

**Fig. 6** PspC1 at division sites protects encapsulated pneumococci from opsonophagocytosis while PspC2 at the pole mediates cell adhesion. **a** Uptake of strain BHN418 or its isogenic *pspC* deletion mutants, with or without human serum, by THP-1 derived macrophages. Graph shows Mean ± SEM of three independent experiments *, $p < 0.05$. **b** Representative immunofluorescence images showing bacteria attached to A549 lung epithelial cells via the poles. Phalloidin stained A549 cells (red) were incubated with FITC stained BHN418 bacteria (green). Scale bar = 5 μm. **c** Quantitative analysis of bacterial adhesion to A549 lung epithelial cells. **d** Adherence of wild type BHN418, or its isogenic *pspC* deletion mutants, to human lung epithelial A549 cells. Graph shows mean ± SEM of four independent experiments. *, $p < 0.05$, **, $p < 0.01$, ***, $p < 0.001$. **e** Schematic presentation of the LPxTG anchoring domain mutant of PspC1. **f** Representative immunofluorescence images of localization of PspC1 and PspC2 on the BHN418*pspC1LpxTG* mutant. Antibody labeled with Alexa fluor 594 (red) was used in combination with mouse anti-PspC1 serum and FITC-labeled secondary antibody (green). Scale bar = 1 μm. **g** Adherence of the BHN418*pspC1LPxTG* mutant to human lung epithelial A549 cells. Graph shows mean ± SEM of three independent experiments. ***, $p < 0.001$

significant reduction in adhesion (Fig. 6d). Thus, the primary role of PspC2 might be to act as a polar adhesin. This adhesive function of PspC2 could also explain the lower uptake of the double mutant BHN418ΔpspC1ΔpspC2 compared to the single mutant BHN418ΔpspC1.

We hypothesized that the localization site of PspC2 relative to the capsular layer within the aging cell wall may allow PspC2 to act both as an adhesin, and to prevent lateral C3b amplification within the wall. As an approach to affect the position of PspC2 within the cell wall we replaced the choline-binding motif of PspC1 with the LPxTG anchoring motif of PspC2 (Fig. 6e). In strain BHN418pspC1LPxTG, expressing both PspC1 and PspC2 as LPxTG proteins, we noted a similar presentation of the cell wall anchored PspC1 at division sites as the choline-binding version of the protein, but a more uniform staining pattern of PspC2 with no preference for the poles, and without clear zones at division sites (Fig. 6f). Whereas C3b binding, FH binding, and opsonophagocytosis were not different from the wt strain BHN418, adhesion to A549 cells was impaired for the mutant, suggesting that the surface presentation of PspC2 molecules relative to the capsule is reduced when a competing protein with the same cell wall sorting sequence is produced by the cell (Fig. 6g).

**Protection against opsonophagocytosis is impaired in a signal peptide switched mutant**. We next asked if we could alter localization and/or surface density of PspC1. Existing literature suggests that localization of surface proteins in Gram positive bacteria can be changed by altering its signal peptide[31]. Although signal peptides of surface proteins in Gram positive bacteria are very conserved in having a YSIRK signal, we found small differences in the sequences between the signal peptides of PspC1 and PspC2 (Supplementary Fig 16). A signal peptide (SP) mutant (BHN418sp2pspC1) was therefore generated by replacing the signal peptide of PspC1 with that of PspC2 (Fig. 7a). STED imaging of BHN418sp2pspC1 showed that PspC1 in most cells was distributed over a larger area of the bacterium compared to in BHN418 (Fig. 7b). We analyzed the distribution pattern of PspC1 by using a signal distribution analyzer and writing a script in MATLAB. The result showed a wider peak in mutant bacteria stained for PspC1 in comparison to wt bacteria where the signal was in the form of sharp peaks over the length of the bacteria (Fig. 7b lower panel). We further compiled fluorescence data from a number of bacterial cells of different lengths. Results for higher order analysis of these STED images show that the fluorescence peaks for the wt cells were narrower/sharper and less spread out than for the mutant (Fig. 7c, Supplementary Fig 17). As seen in Fig. 7C for both wt and the mutant, the smallest cells possessed two PspC1 band rings close to one another. This distance increased with bacterial length. In the longest cells this distance increased to about 1 μm for both wt and mutant bacteria (Fig. 7c). However, due to its broader and more diffuse PspC1 bands the area that lacked fluorescence, between the two bands, was ~30% smaller in the mutant as compared to the wt (Supplementary Fig 18). Thus, PspC1 in the SP mutant retained its localization at division sites but occupied a larger area in the cell wall around these sites.

We then asked whether the altered cell wall distribution of PspC1 in the SP mutant has consequences for bacterial FH binding, and the ability to resist complement-mediated opsonophagocytosis. Even though expression of PspC1 in wt BHN418 and the SP mutant was similar as determined by Western blot analysis (Fig. 7d), the SP mutant exhibited a lower FH binding than BHN418 as determined by flow cytometry analysis (Fig. 7e). By performing double staining of BHN418sp2pspC1 bacteria with

FH and PspC1, FH was shown to bind where PspC1 was expressed as in wt BHN418 (Supplementary Fig 19). However, FH binding showed more narrow bands than PspC1, suggesting that FH is recruited by PspC1 molecules at the division sites where the density of PspC1 is the highest and likely most accessible for incoming FH. The complement deposition on SP mutant bacteria showed a bimodal pattern with a fraction of bacteria with a considerably higher C3b deposition. However, the bulk of bacteria showed similar C3b deposition as wild type BHN418 (Fig. 7f). The SP mutant retained the same resistance to MAC complex formation as BHN418. Importantly, opsonophagocytosis was increased in the SP mutant as compared to in wt BHN418 (Fig. 7g). This was not due to altered presentation and role of PspC2 as a polar adhesin. We therefore suggest that PspC1 needs to recruit FH in sufficient amounts at surface accessible division septa to effectively prevent the building of bulky C3b deposits that otherwise may contact phagocytic C3b receptors on phagocytic cells, enabling phagocytosis.

## Discussion

In this study we show that the coordinated spatial assembly of the pneumococcal cell wall with its associated immuno-protective macromolecules, capsule and FH recruiting proteins, allows pneumococcal evasion against innate immune attack by the human host (Fig. 8). We provide evidence that the pneumococcal capsule assembled via the wzy pathway, appears to be lacking at division septa. Thus, STED microscopy revealed no capsular staining with specific antibodies at the division septa, and SEM images showed a lack of regular surface structures at division septa particularly where cell separation had been initiated. These surface structures were present elsewhere on the cell surface in encapsulated, but completely absent in non-encapsulated isogenic mutants. As the bacterial poles were always covered by capsule, we suggest that the capsule becomes covalently associated or exposed to the outer layers of the peptidoglycan following cell separation.

As a result of no or less capsule at the surface of newly split septal wall, these sites represent the entry port for serum factors, such as C3, across the capsule barrier. Using TEM, deposits of C3b are readily seen at some division septa, but not elsewhere in contrast to in non-encapsulated mutants where C3b becomes deposited all around the cells. Long chains of pneumococci possess higher number of division sites, thus providing more sites for C3b deposition, as shown in the LytB mutant of BHN418 (Supplementary Fig 20), providing a possible explanation for higher complement deposition in bacterial chains compared to diplococci[32].

To evade amplification of C3b, all encapsulated pathogens possess FH binding proteins that inhibit amplification of recruited C3b. We show here that in the pneumococcus, the major FH-recruiting protein PspC, is strategically positioned at each division site. The division site in the pneumococcus is a region in the cell that marks the bacterial cell equator. It was recently discovered that the Mid-cell-anchored protein Z (MapZ) forms ring structures in the membrane at the division site that move apart as new lateral wall becomes inserted between these sites and that MapZ positions the FtsZ ring[8]. As a result the distance between the division site and the bacterial pole remains constant. It is not known whether or not the two division sites each have an elongasome complex locally assembled, producing new lateral wall from each site but in the opposite direction. The localization of PspC at all division sites means that it will become positioned precisely at the equator where the next cell division will become initiated, allowing a spatial closeness between PspC mediated recruitment of FH and C3b deposited at division septa.

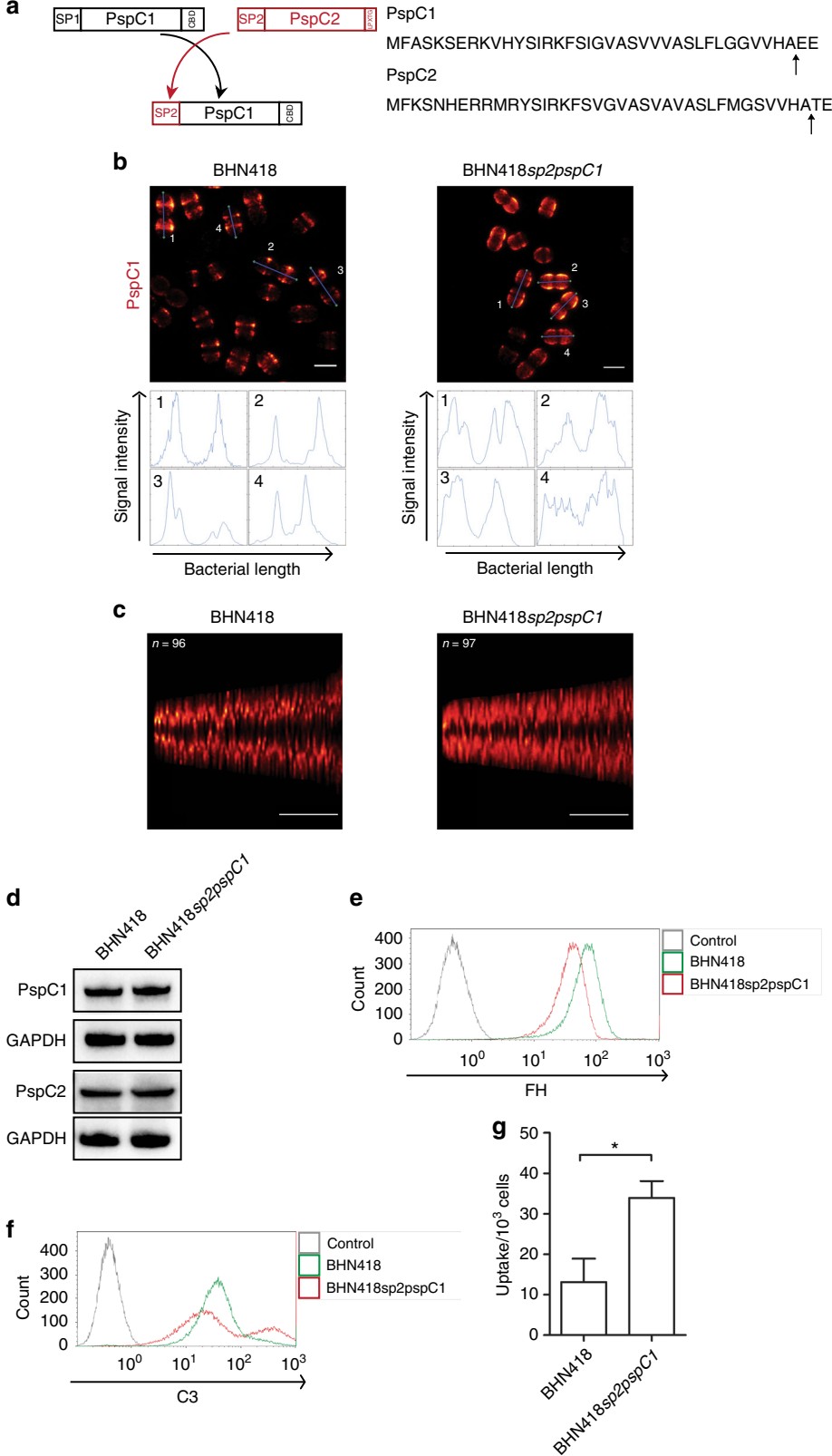

PspC is a choline-binding protein binding choline residues on teichoic acids that either are covalently bound to N-acetylmuramic acid residues on the peptidoglycan (WTA) or to the membrane (LTA)[13]. We do not believe that localization of PspC to division sites is exclusively due to binding to LTA, that like MapZ is membrane bound, since when we replaced the choline-binding motif of PspC1 with the LPxTG motif of PspC2 we still observed PspC1 at division sites. Changing the signal peptide of PspC1 to that of PspC2, having a polar location, did not prevent PspC1 localization around the division sites, but broadened the band which also had negative consequences for FH recruitment.

Super-resolution microscopy using STED reveals for the first time that PspC at division sites is preferentially localized within the wall underneath the capsular layer. This localization of PspC also allows FH to become recruited to the sites in the cell wall where C3b becomes deposited. STED microscopy reveals that in the absence of PspC and FH recruitment to division sites, C3b amplification at division septa can proceed laterally within the wall and underneath the capsule. Appropriate localization of recruited FH may depend on the precise position, within the roughly 36 nm thick cell wall[33], for the FH-binding domain of PspC. This position will depend on if the PspC molecules are anchored to LTA or to WTA and in the latter case if WTA is anchored to a young or old cell wall. Localization of PspC molecules to the outer layers of the peptidoglycan may result in surface exposure, explaining why PspC may also act as a cell adhesin[24].

A small percentage of pneumococci express in addition to PspC also PspC2, a LPxTG protein that we here show is preferentially localized to the bacterial poles and appears to be lacking at division sites. Recombinant PspC1 and PspC2 both bind to FH with similar affinities, but in vivo, PspC2 only marginally contributes to FH recruitment. The majority of PspC2 molecules appear to be localized underneath the capsule, and only recruit FH to the poles when PspC1 at division sites is absent, suggesting that not only C3, but also FH enter encapsulated pneumococci at the division septum, thereby being first recruited by PspC1. Only in the absence of PspC1 FH can diffuse within the wall to PspC2 at the poles. The capsule decreases FH binding in wt bacteria, however, also in capsular mutants, PspC1 contributes more to FH binding than PspC2 (Supplementary Fig 21). We also show that PspC2, that lacks a pIgR motif, acts as an adhesin both to A549 epithelial cells (lacking pIgR) and to Detroit cells (having pIgR), allowing polar attachment to epithelial cells (Supplementary Fig 22). We speculate that at the poles a sufficient number of PspC2 molecules are anchored to the outer layers of the peptidoglycan allowing exposure outside the capsular layer and polar adhesion. This hypothesis is supported by our finding that PspC2 mediated adhesion is completely abolished when

bacterial cells in addition express an LPxTG version of PspC at division sites. Competition for sortase A, a membrane bound enzyme localized near the division septum[34], linking LPxTG proteins to the peptidoglycan[35], may explain this unexpected phenotype.

C3b deposition has two major effects on encapsulated bacteria, promoting opsonophagocytosis by C3b-mediated binding of macrophages, and formation of MAC complexes lysing the cells[25]. We show here that the absence of FH recruitment to division sites increases opsonophagocytosis. We believe that surface exposure of C3b to contact the CR3 receptor, may occur at the division septum. Even laterally amplified C3b in the cell wall underneath the capsule may become exposed at subsequently formed division septa. Also, in the absence of FH recruitment more division septa become decorated by C3b, potentially increasing the number of bacteria accessible to CR3-mediated binding by macrophages. In support of this we note that presentation of PspC over a larger area around the division sites as in the signal peptide switch mutant, results in a much increased opsonophagocytosis. In this mutant a fraction of cells shows an increased C3b deposition compared to the bulk of bacteria, suggesting that it is this fraction, exposing more C3b at division septa, that become oposonophagocytosed.

MAC complex formation in encapsulated pneumococci is a rare event that occurs at or near the division septum. Our super-resolution imaging suggest that MAC complexes are not formed in the wall, as has been proposed for *Streptococcus pyogenes*[26], but in the membrane, as the distance between the C5b-9 signal to the capsular signal seemed to be larger than for C3b and the capsule. Absence of PspC at division sites increased the number of MAC complexes close to division septa, but no spread of multiple MAC complexes on the same cell was observed. The reason why serum killing is usually not observed in encapsulated pneumococci might primarily be due to the low frequency at which MAC complexes are formed.

Our study provides the first evidence of functional and spatial coordination between two major virulence factors, the pneumococcal capsule and FH binding surface proteins. The spatial relationship shown here between the capsule and pneumococcal surface proteins for bacterial complement evasion and adhesive functions may also have implications for the development of new pneumococcal vaccines based on surface proteins, as surface accessibility might be a crucial factor determining their ability to promote antibody-mediated opsonophagocytosis.

## Methods

**Bacterial strains and culture conditions**. Bacterial strains (Supplementary Table 1) were grown overnight on blood agar plates at 37 °C and 5% CO$_2$ and transferred to liquid casitone and yeast extract (C+Y) containing cultures with 5%

---

**Fig. 7** A signal peptide switch mutant of PspC1 affects its surface distribution and immunoprotective function. **a** Schematic presentation of the signal peptide exchange mutant of PspC1 (left panel) and amino acid sequences of the signal peptide of PspC1 and PspC2 (right panel). Arrows represent the signal peptide cleavage site. **b** Representative STED images of the PspC1 distribution in the signal peptide mutant, BHN418sp2pspC1, and in wt BHN 418 bacteria. PspC1 is stained using anti-PspC1 serum and Atto674-N secondary antibody. **c** The fluorescence intensity distribution pattern along the length of bacteria was analyzed using a MATLAB based script. The x-axis represents the individual bacteria with length increasing from left. The y-axis shows the actual length of bacteria (in μm) and the color map represents the fluorescence intensity along the bacteria (with color scale from black via red to yellow for the highest intensities). **b**, **c** Scale bar = 1 μm. **d** Western blot analysis showing the expression level of PspC1 and PspC2 in wt BHN418 and in the signal peptide switch mutant, BHN418sp2pspC1. GAPDH was used as a loading control. Full blots are shown in Supplementary Fig. 23. **e** Representative histogram of FH binding to wt BHN418 and to the signal peptide switch mutant, BHN418sp2pspC1. FH binding was detected using a polyclonal goat anti-FH antibody and a FITC-labeled rabbit anti-goat IgG secondary antibody. Bacteria incubated without FH were used as control. **f** Representative histogram of C3 and C5b-9 deposition on BHN418 and on the signal peptide switch mutant, BHN418sp2pspC1, after incubation with 20% normal human serum. Bacteria were sequentially stained with goat anti-C3 or mouse anti C5b-9 antibody followed by FITC-labeled secondary antibody. **g** Uptake of wt BHN418 or its isogenic signal peptide switch mutant, BHN418sp2pspC1, by THP-1 derived macrophages in the presence of human serum. Graph shows Mean ± SEM of three independent experiments. * = $p < 0.05$. **e**, **f** Representative histogram of three independent experiments

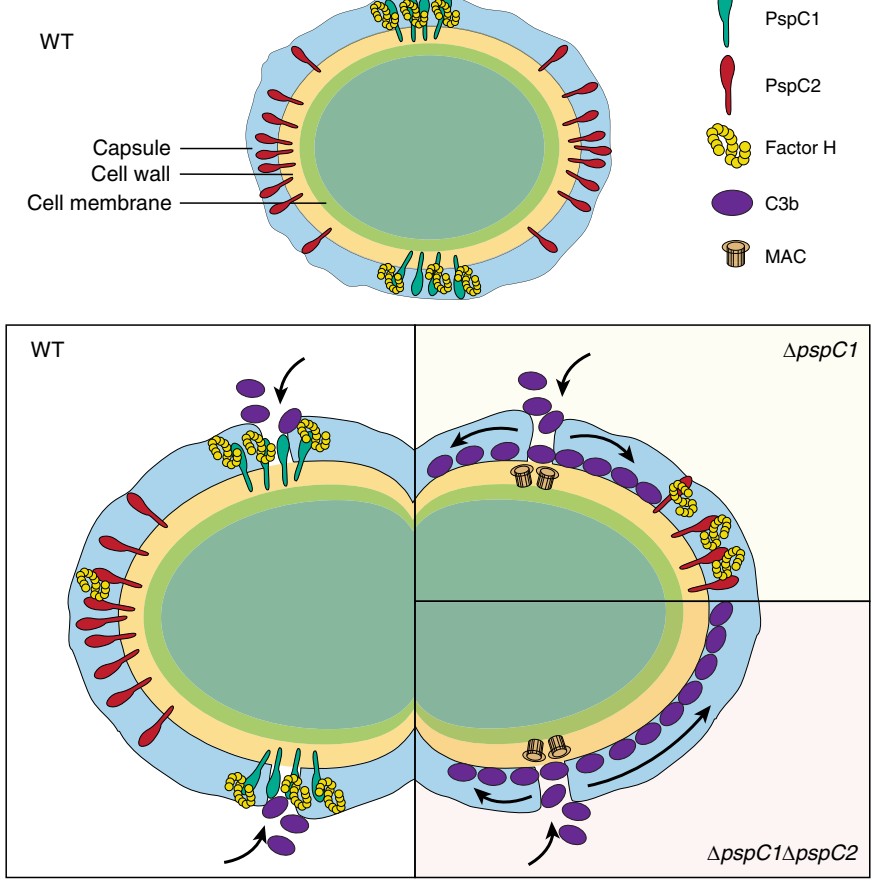

**Fig. 8** Proposed roles for PspC1 and PspC2 in complement deposition and amplification. In wild type (WT) bacteria, PspC1 is presented at the division sites where it recruits human FH. In contrast, PspC2 is presented at the bacterial poles mediating bacterial adhesion to lung epithelial cells. Single cocci (top panel) are more resistant to complement deposition in comparison to diplococci (lower panel) or chains with the latter showing areas of little or no capsule close to the division septum, a site serving as entry port for complement deposition. In the absence of PspC1, FH can bind to PspC2 at the poles and deposited C3b can amplify laterally below the capsular layer except for to the poles, which are protected by PspC2, leading to increased opsonophagocytosis. When both PspC1 and PspC2 are lacking, bacteria are fully covered with C3b including the poles

(wt/vol) dextrose and serum (DS)-containing medium. Unless otherwise stated, strains were grown to mid-log phase (OD620, 0.3–0.4).

**Mutants of *Streptococcus pneumoniae***. All deletion mutants were constructed using polymerase chain reaction (PCR) ligation mutagenesis. PCR amplicons of fragments upstream and downstream of the target gene and the Erythromycin (ermB), kanamycin (kan) cassette, or tetracycline (tet) cassette were generated with specific primer pairs containing overlapping regions. The ermB cassette, kan cassette, or tet cassette was first fused to the upstream fragment and then the downstream fragment was added by PCR. If amplification led to multiple PCR products, the PCR fragment with the correct size was gel-purified after agarose gel electrophoresis. Finally, the fusion PCR product was used to transform *S. pneumoniae*. Mutants were selected on blood agar plates containing erythromycin (1 μg/mL), kanamycin (400 μg/mL) or both. Tetracycline resistance was used to create capsule deletion mutant. The correct insertion was confirmed by PCR and sequencing. The primers used for each mutant are stated in Supplementary Table 3.

**Expression and purification of 6×His-PspC**. PspC (D39), PspC1 (BHN418), and PspC2 (BHN418) were cloned into the NdeI/XhoI or NdeI/BamHI restriction site of pET28a vector (Novagen) and transformed into *E. coli* Rosetta (DE3) cells (Novagen). Recombinant protein was expressed as 6×His-tagged N-terminal fusion protein in logarithmically growing cultures with IPTG (Sigma). Bacteria were resuspended in buffer containing 50 mM Tris·HCl, 50 mM NaCl, and 5% glycerol and disrupted using a Stansted cell disrupter.

6×His-PspC was purified from the soluble fraction by affinity chromatography by using Ni-NTA His Bind resin (Novagen) according to the manufacturer's guidelines. Purified 6×His-PspC2 was incubated rotating with 100 U/mL thrombin (Sigma) for 2 h at room temperature. To remove uncut protein and His-tag, PspC2 was passed over Ni-NTA resin and further purified by size exclusion chromatography by using Superdex 75 gel filtration columns (GE Healthcare). Purity was assessed by electrophoresis on a SDS/PAGE.

**Antibody production directed to PspC2**. Recombinant PspC2 protein lacking the LPxTG motif and the first 37 aa, which constitute a signal peptide region, was expressed and purified in *E. coli* (see above). Purified PspC2 was sent to Innovagen AB for rabbit immunization and production of polyclonal antibodies. Polyclonal anti-PspC2 IgG were purified by protein G affinity chromatography according to the manufacturer's manual using HiTrap protein G-sepharose columns (GE Healthcare).

**Choline chloride treatment of the bacteria**. Bacteria were grown in C+Y media till OD 0.4, washed with PBS and incubated in 140 mM choline chloride solution for 30 min at 37 °C. After PBS wash bacteria were resuspended in PBS for further antibody incubation.

**Peptide mapping**. An array of 48 overlapping peptides was synthesized by Genescript custom peptide synthesis (USA). Each spot contained a 15 amino acids long peptide covalently spotted onto a cellulose membrane with three overlapping amino acids to the next spot and covering the N-terminal domain of PspC1 and PspC2. The peptide array membrane was incubated with PspC1 (a generous gift from Novartis Vaccines, Siena, Italy) and PspC2 antibodies respectively (1:1000) followed by incubation with the HRP conjugated secondary antibody (anti-mouse for PspC1 and anti-rabbit for PspC2) for detection (GE healthcare, NXA931, RPN4301, 1:10,000) (Supplementary Fig 6).

**Factor H binding to pneumococci**. Bacteria were grown at 37 °C in C+Y medium, washed and resuspended in PBS to a concentration of approximately $1 \times 10^8$ bacteria/ml. 100 μl of the bacterial suspension was incubated in 2 μg purified FH (Calbiochem341274,) at 37 °C for 30 min and after three washes, FH binding was detected using a polyclonal goat anti-FH antibody (Calbiochem, 341276, 1:100) and a fluorescein isothiocyanate (FITC)-labeled rabbit anti-goat IgG secondary antibody (Life Technologies, A16143, 1:200). Each incubation step was followed by

three washes in PBS (centrifugation at $1500 \times g$ for 5 min). Bacteria were fixed in 4% paraformaldehyde, washed, and visualized by fluorescence microscopy (Leica or Delta Vision) or analyzed quantitatively by flow cytometry using a Gallios™ flow cytometer (Beckman coulter).

**Surface plasmon resonance/BIACORE analysis of PspC–FH interaction**. The two variants of PspC, PspC1 (BHN418), PspC2 (BHN418), and PspC (D39) were used[22] to analyze PspC–FH interaction by the surface plasmon resonance technique using Biacore X100 equipment (Supplementary Fig 5) (https://www.sprpages.nl/data-fitting/models/kinetic-titration)[36]. Briefly, FH was coupled to the CM4 chip in active and reference flow cell, and recombinant PspCs were injected into the flowcell. Molar ratio for interaction was considered 1:1 for this interaction. Background measurements of the uncoupled flow cell were subtracted. Commercial PBS-P+buffer (GE healthcare) was used for flow. Capture buffer (0.5–1 µg/ml in 10 mM Acetate buffer, pH 4.5) and regeneration buffer (3 M NaCl, 10 mM acetate buffer, pH 4.6) were obtained from GE healthcare.

**Deposition of C3b and C5b-9 (MAC)**. Bacteria were incubated with 20% NHS (Sigma) in PBS for 30 min at 37 °C and washed three times with PBS. Staining was performed sequentially with goat anti-C3 antibody (Calbiochem, 204869, 1:200), recognizing C3b, followed by Alexa fluor 488 labeled Donkey anti-goat antibody diluted (Invitrogen, A11055, 1:200), each incubated for 30 min followed by washing. Bacteria were fixed in 4% paraformaldehyde and visualized by fluorescence microscopy (Leica or Delta Vision) using 100× objective or analyzed quantitatively by flow cytometry using a Gallios™ flow cytometer (Beckman coulter). For C5b-9 staining exponentially growing bacterial cultures were incubated with 20% NHS (Sigma) followed by incubation with anti-C5b-9 antibody (αE11), (Santa cruz biotechnology, sc-58935, 1:50). Alexa fluor 488 labeled secondary antibody (Life Technologies, A11001, 1:100) was used for further detection. Membrane lipids were stained with Nile red (sigma, N3013). After every incubation bacteria were washed twice with 1X HBSS. Samples were fixed and stained with Nile red for 5 min before spreading the samples on glass slides. Samples were visualized using Leica microscope with 100× objective.

**STED imaging**. STED imaging was performed with an instrument from Abberior Instruments (Göttingen, Germany), built on a stand from Olympus (IX83), with a four-mirror beam scanner (Quad scanner, Abberior Instruments), and modified for two-color STED imaging. Two fiber-coupled, pulsed (20 MHz) diode lasers emitting at 637 nm (LDH-D-C, PicoQuant AG, Berlin) and 594 nm (Abberior Instruments) are used for excitation (alternating mode, with the excitation pulses of the two lasers out of phase with each other to minimize cross-talk). The beam of a pulsed fiber laser (MPB, Canada, model PFL-P-30-775-B1R, 775 nm emission, 40 MHz repetition rate, 1,2 ns pulse width, 1,2 W maximum average power, 30 nJ pulse energy) is reshaped by a phase plate (VPP-1c, RPC Photonics) into a donut profile and then used for stimulated emission. The three laser beams are overlapped and then focused by an oil immersion objective (Olympus, UPLSAPO 100XO, NA 1,4) into the sample. The fluorescence is collected through the same objective, separated from the excitation path via a dichroic mirror, passed through a motorized confocal pinhole (MPH16, Thorlabs, set at 50 µm diameter) in the image plane, split by a dichroic mirror and then detected by two single photon counting detectors (Excelitas Technologies, SPCM-AQRH-13), equipped with separate emission filters (FF01-615/20 and FF02-685/40–25, Semrock) and a common IR-filter (FF01-775/SP-25, Semrock) to suppress any scattered light from the STED laser. In this study, a spatial resolution (FWHM) of about 25 nm could be reached. Image acquisition, including laser timing/triggering and detector gating is controlled via an FPGA-card and by the Imspector software (Abberior Instruments). Capsule staining was performed using rabbit anti-capsule serum for serotype 6B, serotype 2 and serotype 4 (Statens Serum Institute, Denmark) and anti-rabbit Atto647N secondary antibody (Sigma, 4039, 1:100).

**STED image analysis**. The projected fluorescence intensity distribution along the main symmetry axis of the bacteria was analyzed by use of higher order moments $\mu_m$, defined as

$$\mu_m = \frac{1}{N} \sum_{k=1}^{N} \left( \frac{I_k - \langle I_k \rangle}{\langle I_k \rangle} \right)^m, \tag{1}$$

where $m = 2,3,4\ldots$ is the order of the moment. $I_k$ is the summed fluorescence intensity over the width of the bacteria, at a pixel $k$ along the length symmetry axis of the bacteria. $\langle I_k \rangle$ is the mean $I_k$, averaged over the whole bacteria, i.e., over all $N$ pixels along the full length axis of the bacteria. If $I_k$ is close to $\langle I_k \rangle$, then $(I_k - \langle I_k \rangle)/\langle I_k \rangle$ will approach zero as the power $m$ increases. This corresponds to the case when the projected intensity trace along the length axis of the bacteria ($I_k$ for $k = 1,\ldots,N$) is more evenly distributed, with only minor deviations from $\langle I_k \rangle$. On the other hand, if the projected intensity trace along the length axis includes high and sharp peaks that markedly deviate from $\langle I_k \rangle$, then $\mu_m$ will not decrease to the same extent with increasing $m$, if at all. Therefore, in bacteria with PspC preferably localized in defined ring structures, in planes perpendicular to the length axis of the bacteria, rather sharp and narrow peaks in the projected intensity traces along these

bacteria are expected. The higher order moments, $\mu_m$, calculated from these bacteria will be markedly higher than those from bacteria where PspC is more evenly spread over the whole bacteria, and we therefore used $\mu_m$ to quantify this difference.

**MATLAB analysis**. Analysis of STED images was carried out using custom written code in MATLAB R2013b. Basically the algorithm worked as follows: In each image bacteria to be analyzed were selected manually by clicking at both ends of the bacteria. MATLAB then calculated a line in between these two points where the line is represented by an array in which each element is the total fluorescent signal integrated orthogonal to the line and over the bacteria's width. In this way the total fluorescence distribution along the length of the bacteria can be stored and analyzed as a fluorescence trace for each selected bacteria. For area estimation, we summed together the patches along the symmetry line where the fluorescence distribution was non-zero. The result was taken in relation to the total length of the bacteria as an estimation of the relative area of the bacteria covered by complement. The fluorescence intensity traces were also ordered after the length of the bacteria in order to create a map of the fluorescence intensity profile for all bacteria, see Fig. 7b and c.

**Co-localization analysis**. The co-localization calculations were carried out using custom written code in MATLAB R2013b. The calculations, analysis and thresholding of images followed the outline described in detail by Xu et al[37]. The co-localization coefficients were based on image cross-correlation spectroscopy (ICCS)[38] as well as Pearsons correlation[39]. By ICCS, co-localization coefficients are obtained for both the red and the green channel. These coefficients are more suited for co-localization analysis when there is an excess of one of the labeled species compared to the other[36]. In the recorded STED images in this study, there was almost always more red than green signal, and we therefore used the amplitude of the ICCS function when correlating the green signal with the red signal (ICCS$_{\text{GREEN}}$), as a measure for co-localization. The ICCS$_{\text{GREEN}}$ coefficient takes on values between 0 and 1, where 0 is no co-localization and 1 is perfect co-localization between the channels. Images were processed by applying a smoothing Gaussian filter in order to reduce noise before the co-localization analysis.

For statistical significance we used a two sided student t-test to test the null-hypotheses that the co-localization coefficients from two different samples have the same mean. This test yields a $p$-value giving the probability for the null-hypothesis being true. If the $p$-value <0.05 the null-hypothesis is rejected with a *significance, with a $p$-value <0.01 the null-hypothesis is rejected with a **significance, and if the $p$-value <0.001 the null-hypothesis is rejected with a ***significance.

**Adhesion assays to human A549 or Detroit 562 cells**. Human A549 cells (lung adenocarcinoma epithelial cell line) (ATCC CCL-185) or human Detroit 562 cells (ATCC CCL-138) were cultured in RPMI 1640 (Invitrogen) supplemented with 10% FBS and 2 mM L-glutamine at 37 °C and 5% CO$_2$. Cells were seeded in 24-well plates ($5 \times 10^5$ cells per well) and grown to confluent monolayers. Cells were washed thoroughly and infected with pneumococci for 1 h in 500 µl RMPI medium without FBS at 37 °C using a multiplicity of infection of 20. After infection, cells were washed extensively to remove unbound bacteria and lysed for 12 min at 37 °C in 1% saponin (Sigma). Serial dilutions were plated on blood agar plates and incubated over night at 37 °C and 5% CO$_2$ to determine the number of adhered bacteria. To assess adhesion by microscopy, A549 cells were seeded in a density of $5 \times 10^5$ per well in a chamber slide and grown to approximately 60% confluency. Cells were further stained with phalloidin (red) while mid-log phase bacteria were stained with FITC (green), and cells were infected with bacteria for 1 h at 37 °C. Samples were fixed with 2% PFA before visualization with microscopy (Delta vision).

**Transmission electron microscopy (TEM)**. *S. pneumoniae* BHN418 or BHN418Δcapsule were grown at 37 °C in C+Y medium until OD620 = 0.15. 20 min post induction cells were centrifuged for 15 min at $5000 \times g$, 4 °C. The pellet was resuspended in 20% NHS (Sigma) and incubated at 37 °C with shaking for 45 min. Bacteria were washed thoroughly and resuspended in 80 µL of phosphate buffered saline (PBS). Drops of 10 µL were placed for 2 min on carbon coated grids (Oxford Instruments, UK). For immunogold staining, grids were fixed with 10 µL of 0.2% glutaraldehyde for 2 min and the reaction was stopped with 10 µL of 1% glycine for 15 min. The grids were then incubated with anti C3 antibody (Calbiochem, USA) diluted 1:100 for 1 h, washed three times with PBS, incubated with donkey anti-goat antibody (18 nm gold particles—Abcam, UK) diluted 1:250 for 1 h. Finally, grids were washed three times with PBS. All grids were negatively stained with 2% uranyl acetate in water. Specimens were examined in a Tecnai 12 Spirit Bio TWIN transmission electron microscope (FEI Company, Eindhoven, Netherlands) operated at 100 kV. Digital images were recorded using a Veleta camera (Olympus Soft Imaging Solutions, GmbH, Münster, Germany).

**Scanning electron microscopy (SEM)**. Bacteria were grown at 37 °C and 5% CO2 in C+Y medium until OD620 = 0.2. They were washed once in PBS and fixed by immersion in 2.5% glutaraldehyde in 0.1 M phosphate buffer (pH 7.4). The specimens were then transferred to a pre-sputtered filter (Polyamide, NL 16, GE Healthcare UK

Limited, Buckinghamshire, UK), rinsed in distilled water and placed in 70% ethanol for 10 min, 95% ethanol for 10 min and absolute ethanol for 15 min all at refrigerator temperature and then into acetone. Specimens were then dried using a critical point dryer (Balzer, CPD 010, Lichtenstein) using carbon dioxide. After drying, filter was mounted on an aluminum stub and coated with Platinum (Q150T ES, West Sussex, UK). The specimens were analyzed in an Ultra 55 field emission scanning electron microscope (Zeiss, Oberkochen, Germany) at 5 kV.

**Western blot**. BHN418 and BHN418sp2pspc1 (Fig. 7d) were grown to mid log phase in C+Y medium and harvested by centrifugation at $3500 \times g$ for 5 min. Bacteria were lysed in a buffer containing 1% triton X100. Ten microgram of protein was used for SDS-PAGE (polyacrylamid electrophoresis) and samples were transferred on Bio-Rad nitrocellulose membrane. After blocking the membrane was incubated with anti PspC1 or anti GAPDH antibody (1:1000). Signal was detected using secondary antibody conjugated with HRP (1:10,000).

For detection of C3d (Supplementary Fig 14), BHN418 and isogenic *pspC* mutants were grown to mid log phase in C+Y medium and harvested by centrifugation at $3500 \times g$ for 5 min. Bacteria were incubated with 20% NHS (Sigma) in PBS for 60 min at 37 °C and washed three times with PBS. Ten microgram of protein was loaded on SDS-PAGE and samples were transferred on Bio-Rad nitrocellulose membrane. After blocking the membrane was incubated with anti-C3d antibody (Abcam,ab17453,1:2000). Signal was detected using secondary antibody conjugated with HRP (1:10,000).

For detection of the specificity of anti PspC1 and anti PspC2 antibody (Supplementary Fig 7), BHN418 was grown to mid log phase in C+Y medium and harvested by centrifugation at $3500 \times g$ for 5 min. Bacteria were lysed in a buffer containing 1% triton X100. Ten microgram of protein was loaded on SDS-PAGE and samples were transferred on Bio-Rad nitrocellulose membrane. After blocking the membrane was incubated with mouse anti-PspC1 or rabbit anti-PspC2 antibody (1:4000). Signal was detected using secondary antibody conjugated with HRP (GE healthcare) (1:10,000).

**Phagocytosis assay**. Human monocytic leukemia THP-1 cells (ATCC TIB-202, Manassas, VA) were cultivated in RPMI 1640 (Invitrogen) supplemented with 10% FBS and 2 mM L-glutamine at 37 °C and 5% $CO_2$. THP-1 cells were seeded in 24-well plates ($5 \times 10^5$ cells per well) and differentiated for 48 h with 20 ng/ml of phorbol myristate acetate (PMA) (Sigma). Bacteria were incubated with 20% NHS (Sigma) in PBS for 30 min at 37 °C for opsonization and washed three times with PBS. Cells were washed with PBS and infected at a MOI of 60 with opsonized or non-opsonized pneumococci and resuspended in RMPI medium without FBS. After 1.5 h of incubation, cells were washed three times with PBS, and incubated for a further 1 h in the presence of 300 μg/ml gentamycin to kill extracellular bacteria. The cells were subsequently washed three times with PBS and lysed for 12 min at 37 °C with 1% saponin in PBS. Serial dilutions were plated on blood agar plates and incubated over night at 37 °C and 5% $CO_2$ to determine the number of phagocytosed bacteria.

For phagocytosis assay in the presence of pure FH and pure C3b, bacteria were grown at 37 °C in C+Y medium, washed and resuspended in PBS to a concentration of approximately $1 \times 10^8$ bacteria/ml. 200 μl of the bacterial suspension was incubated in either 2 μg purified FH (Calbiochem) or 2 μg purified C3b (Calbiochem, 204860) at 37 °C for 30 min and after three washes with PBS, bacteria were used in the uptake assay described above.

**Statistical analysis**. Results are expressed as mean ± SEM or SD. Statistical significance was assessed using one-way Anova and Bonferroni posttest unless otherwise specified. All analyses were performed using GraphPad Prism® version 5.04 or Python documentation. $P$-values <0.05 were considered significant.

**Code availability**. The custom codes used in this study are available at https://github.com/janjeb/Pneumucocci_Matlab.

**Data availability**. The data that support the findings of this study are available in this article and its Supplementary Information files, or from the corresponding author upon reasonable request.

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

## Acknowledgements

The authors thank Novartis for providing Anti-PspC antibody, Ewa Idsund Jonsson and Kjell Hultenby at the Electron microscopy unit (EMil), Karolinska University Hospital Huddinge for electron microscopy, Åsa Frostell from GE healthcare for her support in performing BIACORE analysis. Staffan Normark, Tim Schulte, Adnane Achour, Peter Mellroth and Martin Norman for helpful discussions. This work was supported by grants from the Swedish Research Council, ALF grant from Stockholm County Council, the Swedish Foundation for Strategic Research (SSF), and Knut and Alice Wallenberg foundation.

## Author contributions

A.P, B.H.N., and JW designed the study. A.P., V.S., L.S., J.B., M.S.A., and S.M. performed experiments. A.P. and B.H.N. wrote the manuscript. All authors contributed to the writing.

## Additional information

**Competing interests:** The authors declare no competing interests.

