## [Peer Review File · Nature Communications]

Reviewers' comments:

Reviewer #1 (Remarks to the Author):

The manuscript by Anuj Praha describes an interesting high-resolution localization of pneumococcal factor h binding proteins at the division septa of encapsulated *S. pneumoniae*. The authors use a strain that expresses two different PspC variants of strain BHN418. They demonstrate different localization of the two proteins at the surface of pneumococci. The data are interesting and have the potential to elucidate a new role of pneumococcal PspC variants in immune evasion of *S. pneumoniae*.

The authors use super resolution microscopy and localize the site where in intact *S. pneumoniae* bind factor h, where PspC variants are expressed and where c3 fragments are deposited. With this imaging technique they reveal a very precise location of the immune evasion proteins and of deposited c3 fragments. This provides novel aspects but also leaves some open remaining issues.

The authors show c3 fragment deposition at specific sites of the bacterial surface when the bacteria were incubated in 20 % NHS. In addition they demonstrate factor h binding. Here however in this case purified factor h is used. How is factor h staining when bacteria are incubated in 20 % NHS. This patterns must be shown for the three strains analyzed and for the PspC mutants that were generated.

The authors used high resolution microscopy to localize one of the pneumococcal factor h binding proteins PspC1 in the division septum of the bacterium. The second PspC2 protein has a different localization at the bacterial poles. Suggesting a different surface distribution and different roles for immune evasion.

What is not addressed in this study is why then both PspC variants bind factor h in vitro, but upon incubation with purified proteins, factor h binds primarily to the side where PspC1 is expressed. Is this due a protective shielding effect of the capsule but then later on PspC2 seem s to have a role in adhesion.

Upon removal of choline anchored variant factor H binding is also mediated by PspC2. What is the reason for this change in binding? Apparently also PspC2 binds factor h.

Figure 2: The staining pattern for factor h in the BHN418 Δ pspC1 mutant seems also localized to the septum. Based on the images provided. How do the authors explain this phenomenon in case of the different distribution of PspC2.

Why is cholinchloride treatment performed with a different strain, i.e. BHN191 and not with the strain BHN418 and the three mutants?

The authors need to demonstrate what kind of c3 fragment is deposited at the bacterial surface. Apparently factor h assists in the inactivation of c3b in presence of a cofactor factor i. So it is of interest to follow if deposited c3b is degraded and processed to other fragments like c3d and c3dg.

What is the explanation for the rather restricted occurrence of c5b-9 deposition. Is the blockade at most septa of complement cascade so efficient that the cascade does not progress, or do the bacteria have additional complement evasions proteins that block C5b-9 formation?

Is PspC the only factor H binding protein of *S. pneumoniae*. The authors report on the existence of several complement binding proteins in *S. pneumoniae*. However it appears from the data in Figure 4a that PspC is the only factor h binding protein.

The authors show different roles for PspC1 and PspC2 in strain BHN418. Most *S. pneumoniae* express only one PspC variant. How do these other pneumococci deal with septum control and adhesion control. Or how relevant are the findings for other *S pneumonia* strains that express one PspC variant.

PspC is a highly diverse and multifunctional protein. Are all PspC variants expressed at this specific site s at the division septum?

The authors test opsonophagocytosis. Deletion of PspC1, but not of the PspC2 gene enhances opsonophagocytosis. If PspC1 and c3b deposition occurs at the septum below the capsule as shown in the previous experiments, how can c3b or the c3 fragments be accessible for phagocytosis receptors.

Can the authors exclude that PspC proteins bind other human plasma proteins that may have a role in both adherence or phagocytosis ?

The authors mention factor h binding proteins in complement evasion of *Streptococcus pneumoniae*. However their work focuses on one factor h binding protein, PspC.

C3b deposition and C5b-9 deposition are not in parallel. What is the reason and what level of complement control does occur.

Do they correlate with the density of PspC variants

Are other *S pneumoniae* complement evasion proteins involved in this down regulation of the complement terminal pathway.

Figure 4: How do the authors explain factor h binding to the BHN418 Δ pspC1 mutant. Is the surface different or the capsule differently accessible?

Why was another strain BHN191 used for treatment with cholinechloride. What is the binding in the untreated strain and which PspC variant is expressed by this strain?

It is essential to do the same treatment with wild type BHN418 and the BHN418 Δ pspC1 BHN418 Δ pspC2 mutant strains.o

The different factor h binding patterns to the BHN418 Δ pspC1 suggests that the structure is different in this strain and under these conditions pspC2 has a role in factor h recruitment and immune evasion. This situation also can be used to evaluate if other factor h binding proteins are expressed by the pathogen.

Apparently strain BHN418 has specific features and for this strain PspC1 variant is essential for recruitment of factor h. For this scenario the authors nicely demonstrate using super resolution microscopy a specific localization of the PspC1 protein and a unique role.

Exchange of signal peptides. Signal peptides of PspC variants are highly conserved. What which residues are different in the two signal peptides. Thus indicating that residues in the signal peptide direct the expression pattern of the protein. How does this relate to other PspC variants and for strains that express just one single PspC variant.

Have the authors considered that factor h is also discussed to interact and bind to CR3 adhesion

receptor?

Minor

Introduction. C3 is not a complement factor

The specificity of the new antisera generated should be shown

For Supplementary Table S2: the Bicaore experiments the real binding data and the langmuir binding models would be informative. Thus the fitting parameters of the experiment should be included and the concentration range tested.

For Supplementary Figure S2: in panel A what peptide spot represents PspC1 and which PspC2
Be careful with a generalization of linear peptide binding motifs as binding residues.

Please be consistent in using C5b-9 vs MAC (e.g. page 10, line 290)

Reviewer #2 (Remarks to the Author):

Here, the authors use STED microscopy to study the spatial distribution of C3 (and related proteins) in Bacteria. My expertise is not in the biology so I can't comment with authority on the novelty, but I havent seen similar studies and so it seems a nice advance.

On the imaging side, while the microscopy itself seems quite well done (and quite well described), there is a worrying lack of quantification throughout (other than some basic cell counting). If this were done (and of course, if the results hold up) I would consider this manuscript acceptable for publication. The corrections necessary are quite extensive however.

Most notably is a lack of rigorous co-localisation analysis. This require the calculation of the Pearson's or Mander's colocalisation coefficients for each image and then performing statistics| t-tests to test for significant differences between those coefficients in different conditions.

Additionally - the authors describe sub-cellular localisation quite informally. It should be possible to quantify how "even" a staining pattern is and (again) calculate significances between conditions. This might involve the use of additional stains to highlight specific cellular organelles/regions and then test for degrees of colocalisation. Statements about cell shape and morphology can also be quantified.

The final important one is the staining intensities - describes as "low" for example. These should be quantified and (since the staining intensity is fairly arbitrary) probably be given as fold changes (with significances) between conditions.

Finally, the authors should be careful with Figure formatting e.g. some axes are small and quite hard to read e.g. 5b,d.

The M&M says a Leica microscope with 100X resolution was used. this is the magnification not the resolution.

Reviewer #3 (Remarks to the Author):

The study aims to elucidate if the immune defence mechanisms exerted by the capsular polysaccharide (CPS) of pneumococci and the Factor H binding proteins (FHBPs), are spatially coordinated on the bacterial cell surface. The CPS protects against phagocytosis and FHBPs prevent binding of C3b deposition, which otherwise would activate the complement cascade. Here the authors have analysed deposition of C3b and Factor H binding in wild type pneumococci, strains with capsule, and in capsule lacking pneumococci. The study is unique and very interesting because host protein binding was visualized by CLSM and STED, supported by classical binding studies. In addition, two types of PspC proteins, named PspC1 and PspC2, are produced by the selected strain BHN418 (a serotype 6B) and importantly, the mechanism of cell wall anchoring is different for both. While PspC1 is a choline binding protein, PspC2 is anchored by the sortase. The study now figured out when and where PspC1 or PspC2 are located at the bacterial cell surface (in the wild type vs unencapsulated strain) to be accessible for the interaction with Factor H (thereby avoiding C3b deposition). The results demonstrate nicely 1. that the distribution of PspC1 and PspC2 on the surface of wild type pneumococci is different and 2. that PspC1 is located at the division sites, while PspC2 is located at the poles or distributed among the cell surface, similar to RrgB (pilus-1 backbone). These findings are quite surprising and new, because none of the former studies has illustrated the distribution of PspC molecules (or other choline-binding proteins) on the bacterial cell surface.

This PspC localization in CPS containing bacteria has consequences for the interaction with Factor H and deposition of C3b as well. C3b and Factor H binding was detected in regions lacking CPS, hence in the septa of dividing cells, where only PspC1 is located. Thus, the TCC (C5b-C9) was also only detected in the division septa of a few bacteria.

In consequence the findings show that the mechanism of immune evasion by the CPS and PspC1 or PspC2 is a coordinated process, and that PspC1 is probably masked by the CPS but exposed at the division sites to prevent attack by C3b and the MAC.

In general the study is highly interesting and provides new insights into the mechanistic functions of a unique class of surface protein. The distribution and exerted function(s) strongly depend on the anchoring motif. However, the study lacks several details regarding the applied methods and host cells used, and the reviewer has also some concerns about the specificity of tools used and methods applied to elucidate the PspC functions.

Specific comments:

1. The summary is not very informative because it contains mostly state of the art knowledge. The breaking news are not mentioned clearly and this makes it difficult to understand the importance of the results
2. In general the titles in the results section are not well defined. More specific titles indicating the major findings have to be given
3. C3b deposition: the TEM in Figure 1 shows dark regions, probably of C3b deposition. Here immunogold and appropriate controls are the better choice
4. How can C3b be deposited on the cell surface and detected by antibodies, when Factor H is also bound and inactivates C3b into C3bi?

5. The authors used polyclonal antibodies raised against PspC1 and PspC2 and used these antibodies to differentiate between PspC1 and PspC2. According to the document and the partial sequence provided by PspC types it is not clear at all why there is a specific binding of the antibodies to PspC1 or PspC2. An additional experimental evidence has to be given to demonstrate protein specificity. Line 146 mentioned that PspC2 belongs to the TIGR4 PspC, which has (according to the performed sequence alignment done by the reviewer) a high homology to the D39 PspC (these are orthologues!).

6. In line with this, the total sequences of PspC1 and PspC2 of BHN418 has to be provided in the supplement as well as a sequence alignment of the N-terminal regions

7. PspC proteins have a modular organization with different function domains – more information a needed for non-experts on pneumococci and PspC to understand the molecules and the findings by Pathak et al

8. Fig. 4: the authors constructed pspC mutants and showed binding of Factor H only to pneumococci expressing PspC1 in the histograms. However, the immunofluorescence images show binding also to PspC2. Please explain this discrepancy

9. A major question for PspC1 and PspC2 remains the killing of pneumococci by formation of the MAC in wild type and mutant pneumococci (bactericidal assay).

10. Opsonophagocytosis assays: the results of the opsonophagocytosis assay are not clear. As a result of serum incubation pneumococci bind several components of the serum. In wild type CPS expressing bacteria they probably bind Factor H and other components. Factor H is known to recognize integrins on phagocytes and thus, enhanced uptake by e.g. PMNs has been described. In the view of the reviewer the effects of Factor H and the other serum components has not been considered. How is phagocytosis affected in the presence of C3b alone or Factor H alone?

11. When the reviewer is correct, both PspC molecules possess the binding domain for the polymeric Ig receptor. However, this receptor is not expressed by A549. Therefore, the results of this paragraph are unclear to the reviewer, because neither the PspC-pIgR interaction nor the Factor H mediated adherence was investigated. Alternatively, why is there a dramatic effect on adherence when the appropriate receptor for PspC is missing?

12. Figure 7 lacks the immunoblot for PspC2

13. Page 12: the first part of the discussion is a repetition of the introduction. These parts can be combined

Minor suggestions

- Not all choline binding proteins contain the choline binding motifs at the carboxy-terminal part
- The study lacks important citations and findings of original studies. PspC is, according to the literature, a multifunctional protein and divided into different subclasses. In addition to bind Factor H, here with prototypes of the "conventional" PspC and the sortase-anchored Hic, both types have been shown to bind vitronectin, which is also a complement inhibitor. Binding studies by SPR even analysed in depth the protein-protein interactions
- The binding regions (not the motifs) have been described for PspC and Hic. The simultaneous binding or even the competitions between Factor H and vitronectin binding or sIgA binding have also been reported. These facts are important to understand the function of PspC/Hic in invasive infections

- In addition, PspC (but not the Hic subtypes) link *S. pneumoniae* with the pIgR (several publications). The consequences of this interaction or with free sIgA has been reported
- The above mentioned findings are not referenced in the Introduction, but represent an important aspect when incubating pneumococci with serum.
- It is therefore not possible to restrict the analysis to Factor H binding when incubating pneumococci with serum
- The sensorgrams of the SPR studies have to be shown and the evaluation of the data has to be provided. All details of the SPR experiments are missing (coupling procedure, analyte concentration and flow...)
- Line 330: correct the sentence "We show here show..."
- Line 342: PspC is not decorating choline residues on teichoic acids
- In general: check the references and citations
- Are pneumococci killed by using only gentamicin?

Point-by-point response to the comments made by the reviewers

Reviewer#1

The manuscript by Anuj Praha describes an interesting high-resolution localization of pneumococcal factor h binding proteins at the division septa of encapsulated S. pneumoniae. The authors use a strain that expresses two different PspC variants of strain BHN418. They demonstrate different localization of the two proteins at the surface of pneumococci. The data are interesting and have the potential to elucidate a new role of pneumococcal PspC variants in immune evasion of S. pneumoniae.

We thank the reviewer for this comment.

The authors use super resolution microscopy and localize the site where in intact S. pneumoniae bind factor h, where PspC variants are expressed and where c3 fragments are deposited. With this imaging technique they reveal a very precise location of the immune evasion proteins and of deposited c3 fragments. This provides novel aspects but also leaves some open remaining issues.

The authors show c3 fragment deposition at specific sites of the bacterial surface when the bacteria were incubated in 20 % NHS. In addition they demonstrate factor h binding. Here however in this case purified factor h is used. How is factor h staining when bacteria are incubated in 20 % NHS. This patterns must be shown for the three strains analyzed and for the PspC mutants that were generated.

Response: To address this concern by the reviewer, we have included new figures in **Suppl. Fig. S10**, illustrating the FH deposition pattern on wild type TIGR4, D39, and BHN418 as well as for the *pspC* deletion mutants. FH deposition shows similar pattern on the bacterial surface when 20% NHS is used as a FH source as compared to when purified FH is used.

Suppl. Fig. S10. Localization of factor H (FH) on different pneumococcal strains incubated with 20% NHS. Representative immunofluorescence images of FH deposition on strain (a) BHN418 and its isogenic (b) *pspC1* and (c) *pspC2* mutants, as well as on strains (d) TIGR4 and (e) D39. The strains were incubated with 20% NHS, and FH was detected using a polyclonal goat anti-FH antibody and a FITC-labeled rabbit anti-goat IgG secondary antibody (green).

The authors used high resolution microscopy to localize one of the pneumococcal factor h binding proteins PspC1 in the division septum of the bacterium. The second PspC2 protein has a different localization at the bacterial poles. Suggesting a different surface distribution and different roles for immune evasion.

What is not addressed in this study is why then both PspC variants bind factor h in vitro, but upon incubation with purified proteins, factor h binds primarily to the side where PspC1 is expressed. Is this due a protective shielding effect of the capsule but then later on PspC2 seems to have a role in adhesion. Upon removal of choline anchored variant factor H binding is also mediated by PspC2. What is the reason for this change in binding? Apparently also PspC2 binds factor h.

Response: As suggested by the reviewer we investigated the possible protective shielding effect of the capsule. We generated capsular deletion mutants in wild type BHN418 and in its

isogenic *pspC* mutants. The capsular deletion mutant in BHN418 showed higher FH binding in comparison to the encapsulated strain suggesting a shielding effect of the capsule in FH recruitment (new **Suppl. Fig. S19a**). Non-encapsulated *pspC* mutants were then compared for FH binding. We observed that the *pspC1* deletion mutant was more compromised in FH binding than the *pspC2* deletion mutant, suggesting a minor role for PspC2 in FH recruitment even in the absence of the capsule (new **Suppl. Fig. S19b**).

Also, protective shielding by the capsule in FH recruitment has been suggested in a previous study by Hammerschmidt et. al. (*J. Immunol.*, 2007;178 (9):5848-5858).

It is important to note that the purified recombinant proteins that were used in the FH binding experiments lack the respective Choline binding domain or LPxTG domain.

Suppl. Figure S19. The pneumococcal capsule affects FH binding. Representative histogram of FH binding to (a) wt BHN and its isogenic non-encapsulated mutant as well as (b) its isogenic *pspC* deletion mutants using flow cytometry. Bacteria were incubated with purified human FH and stained as in Fig. 4a. Bacteria incubated without FH were used as a control for each strain. The histogram shown is representative of three independent experiments.

a

b

Figure 2: The staining pattern for factor h in the BHN418 Δ pspC1 mutant seems also localized to the septum. Based on the images provided. How do the authors explain this phenomenon in case of the different distribution of PspC2.

Response: Figure 2 shows localization of C3b and C5-9. However, in Figure 4, FH binding is shown for the mutant BHN418 Δ pspC1. FH is mainly observed at the poles where also PspC2 is localized in diplococci and in chains of this mutant. In a long chain septum separation can be at different stages. At late stages they are close to represent the new cell poles likely explaining the localization of PspC2 and FH to such late constrictions in the *pspC1* mutant (Figure 4d).

Fig. 4d, The deletion mutant of PspC1, BHN418 Δ pspC1, was incubated with purified human FH, and FH was detected using a polyclonal goat anti-FH antibody and a FITC-labeled rabbit anti-goat IgG secondary antibody (green). Bacteria were then stained for PspC2 using serum purified rabbit anti-PspC2 polyclonal antibody labeled with Alexa fluor 594 (red).

Why is cholinchloride treatment performed with a different strain, i.e. BHN191 and not with the strain BHN418 and the three mutants?

Response: We could not generate a mutant in strain BHN191 (of serotype 6B and CC138), as this strain was found to be non-transformable. Therefore we performed choline chloride treatment to remove the choline binding protein PspC1 in strain BHN191. We now, as suggested by the referee also performed choline chloride treatment using strain BHN418 and the *pspC* mutants, showing similar results as for strain BHN191 (new Figure 4e and new Suppl. Fig. S9). There was no detectable FH staining when BHN418 Δ pspC2 and BHN418 Δ pspC1 Δ pspC2 were treated with choline chloride (data not shown).

Fig. 4e. Choline chloride treated BHN418 bacteria were incubated with purified human FH, and FH was detected using a polyclonal goat anti-FH antibody and a FITC-labeled rabbit anti-goat IgG secondary antibody (green). Bacteria were then stained for PspC2 using serum purified rabbit anti-PspC2 polyclonal antibody labeled with Alexa fluor 594 (red).

Suppl. Fig. S9. Localization of FH and PspC2 after choline chloride treatment of the pneumococcal strain BHN191, and of *pspC* mutants in wt BHN418.

Representative immunofluorescence images of PspC2 localization in choline chloride treated strains: (a) BHN191, belonging to the same serotype /6B) and sequence type (CC138) as BHN418, and of (b) the isogenic mutant in *pspC1* in wt BHN418, BHN418 Δ *pspC1*. Bacteria were incubated with purified human FH. FH and PspC2 were stained as in Fig. 4c. There was no detectable FH staining when BHN418 Δ *pspC2* or BHN418 Δ *pspC1* Δ *pspC2* were treated with choline chloride (data not shown).

a

b

The authors need to demonstrate what kind of *c3* fragment is deposited at the bacterial surface. Apparently factor *h* assists in the inactivation of *c3b* in presence of a cofactor factor *i*. So it is of interest to follow if deposited *c3b* is degraded and processed to other fragments like *c3d* and 5487/45*c3dg*.

Response: In agreement with the reviewer's comment we analyzed deposited complement components on wild type BHN418 and its isogenic PspC mutants after incubation of the bacteria with 20% NHS using western blotting. We detected C3d fragments using antiC3d antibody in wild type BHN418 and in the mutant BHN418 Δ *pspC2*, but BHN418 Δ *pspC1* and BHN418 Δ *pspC1* Δ *pspC2* showed very faint bands of C3d (new **Suppl. Fig. S13**).

Suppl. Figure S13. Detection of the degradation product C3d

Western blot analysis showing deposited C3d in wt strain BHN418 and in its isogenic *pspC* mutants, BHN418 Δ *pspC1*, BHN418 Δ *pspC2* and BHN418 Δ *pspC1* Δ *pspC2*.

What is the explanation for the rather restricted occurrence of *c5b-9* deposition. Is the blockade at most septa of complement cascade so efficient that the cascade does not progress, or do the bacteria have additional complement evasion proteins that block *C5b-9* formation?

Response: We agree with the reviewer that there is a restricted occurrence of *C5b-9* deposition in wild type bacteria. *C5b-9* is the terminal product of the complement cascade, which requires longer stable complement deposition. Probably blockade at most septa must be so efficient that the cascade does not progress. We cannot exclude the possibility of additional complement evasion proteins, but we observed more *C5b-9* deposition in the PspC1 deletion mutant than in the wild type BHN418 strain (**Figure 5d**), suggesting a major role played by PspC1 in complement evasion.

Fig. 5d. Representative histogram of *C5b-9* deposition on strain BHN418 and on its isogenic *pspC* deletion mutants after incubation with 20% normal human serum using flow cytometry. Bacteria were sequentially stained with mouse anti-*C5b-9* antibody followed by anti-mouse Alexa fluor 488 labeled secondary antibody.

Is PspC the only factor H binding protein of S. pneumoniae. The authors report on the existence of several complement binding proteins in S. pneumoniae. However it appears from the data in Figure 4a that PspC is the only factor h binding protein

Response: Other authors have reported that there are other FH binding proteins in pneumococci for example the Tuf protein (Mohan, S. et al. *Mol Immunol* 2014;62, 249-264). However, in our experimental setting we could not detect any FH binding in the absence of PspC1 and PspC2 in encapsulated pneumococci, suggesting that there is very little or undetectable binding by other FH binding proteins. We did though find FH binding in the double mutant (*pspC1* and *pspC2*) in the absence of the capsule (new **Suppl. Fig. S19b**, see response above), which might suggest that additional FH binding proteins are exposed when the capsule is absent.

The authors show different roles for PspC1 and PspC2 in strain BHN418. Most S. pneumoniae express only one PspC variant. How do these other pneumococci deal with septum control and adhesion control. Or how relevant are the findings for other S pneumonia strains that express one PspC variant. PspC is a highly diverse and multifunctional protein. Are all PspC variants expressed at this specific site s at the division septum?

Response: In agreement with the comment by the reviewer we show that the PspC localization pattern is similar in two other pneumococcal strains: TIGR4 of serotype 4 and strain D39 of serotype 2 (**Figure 3c**). These two strains only express one PspC variant, and different PspC variants. Other researchers have shown that PspC in TIGR4 and D39 can act as an adhesin, but there are also other adhesins present in these strains.

Fig. 3c. Representative immunofluorescence images of PspC localization on strains TIGR4, D39 and BHN418. PspC was detected with mouse anti-PspC antiserum (D39) followed by anti-mouse Alexa fluor 488 (green) secondary antibody.

The authors test opsonophagocytosis. Deletion of PspC1, but not of the PspC2 gene enhances opsonophagocytosis. If PspC1 and c3b deposition occurs at the septum below the capsule as shown in the previous experiments, how can c3b or the c3 fragments be accessible for phagocytosis receptors.

Response: In Figure 1b we observe that C3b is seen as a bulky deposits on the bacterial surface (encapsulated BHN418), which probably makes C3b available for the phagocytic receptors. In addition, we have performed TEM analyses using immunogold labelling of C3b on both encapsulated and non-encapsulated BHN418 and we observe gold particles at or close to division septa in encapsulated strains (new **Suppl. Figure S1**).

Suppl. Fig. S1. TEM images of immunogold staining of C3b on bacteria. Representative TEM images of immunogold staining of C3b in wt BHN418 and non-encapsulated BHN418 after incubation with 20% normal human serum. Zoomed part of images are shown for better visibility of gold particles.

BHN418

BHN418Δcapsule

Can the authors exclude that PspC proteins bind other human plasma proteins that may have a role in both adherence or phagocytosis ?

Response: We cannot exclude the possibility that the PspC proteins can bind other host components such as vitronectin and play an indirect role in adhesion to epithelial cells.

The authors mention factor h binding proteins in complement evasion of Streptococcus pneumoniae. However their work focuses on one factor h binding protein, PspC.

Response: Since we study two PspC variants, PspC1 and PspC2, found in our clinical isolate BHN418, we stated FH binding proteins.

*C3b deposition and C5b-9 deposition are not in parallel. What is the reason and what level of complement control does occur. Do they correlate with the density of PspC variants. Are other *S pneumoniae* complement evasion proteins involved in this down regulation of the complement terminal pathway*

Response: Since C5b-9 is the terminal product of the complement cascade, C5b-9 is formed at places where the complement deposition is high, and probably expected to stay longer on the bacterial surface. This results in less C5-9 than C3b deposition. There are other pneumococcal proteins on the bacterial surface like PspA, which have been reported to inhibit C3b deposition by controlling factor B mediating the alternative complement pathway (*Tu et al, Infect. Immun. 1999; 67: 4720-4724*). Therefore we cannot exclude a possible role of other proteins in inhibition of C3b deposition. However, our data suggest that the PspC proteins are the major contributors to the inhibition of complement deposition.

How do the authors explain factor h binding to the BHN418 Δ pspC1 mutant. Is the surface different or the capsule differently accessible?

Response: We suggest that FH binds to PspC2 in the absence of PspC1 since we found that FH binds differently in the BHN Δ pspC1 mutant as compared to the wild type strain, and that FH co-localizes at the poles with PspC2 on the bacterial surface in this mutant. Also we know that PspC2 can bind FH at least *in vitro*.

Why was another strain BHN191 used for treatment with cholinechloride. What is the binding in the untreated strain and which PspC variant is expressed by this strain? It is essential to do the same treatment with wild type BHN418 and the BHN418 Δ pspC1 BHN418 Δ pspC2 mutant strains.

Response: Since we were not able to generate mutants in BHN191 due to the fact that this strain is non-transformable, we used choline chloride treatment to remove the choline binding protein PspC1 in this strain. Both strains BHN191 and BHN418 have two PspC proteins, PspC1 and PspC2, and they are highly genetically related. As suggested by the reviewer we now performed choline chloride treatment also of BHN418 and its mutants (see response above, new **Figure 4e**, and new **Suppl. Fig. 9**) showing similar results as seen for BHN191.

The different factor h binding patterns to the BHN418 Δ pspC1 suggests that the structure is different in this strain and under these conditions pspC2 has a role in factor h recruitment and immune evasion. This situation also can be used to evaluate if other factor h binding proteins are expressed by the pathogen. Apparently strain BHN418 has specific features and for this strain PspC1 variant is essential for recruitment of factor h. For this scenario the authors nicely demonstrate using super resolution microscopy a specific localization of the PspC1 protein and a unique role.

Response: We agree with the reviewer. Also we detected almost no FH binding in the double mutant in PspC, BHN418 Δ pspC1 Δ pspC2, suggesting that PspC1 and PspC2 are the major FH binding proteins in encapsulated BHN418.

Exchange of signal peptides. Signal peptides of PspC variants are highly conserved. What residues are different in the two signal peptides. Thus indicating that residues in the signal peptide direct the expression pattern of the protein. How does this relate to other PspC variants and for strains that express just one single PspC variant.

Response: We have inserted a signal peptide sequence alignment in the supplement (new **Suppl. Fig. S15**) showing different alleles of PspC. We find that most of the sequences in the different alleles of PspC are conserved including a conserved YSIRK motif. However, there are small differences such as the presence of a twin arginine Motif in the signal peptide of PspC2 (new **Suppl. Fig. S3, S4, S6**). Although it has not been reported so far that there exist a Twin arginine translocation pathway in *Streptococcus pneumoniae*, we predict a possible mechanism for differential protein export with signal peptide sequences carrying twin arginine motifs.

Suppl. Fig S15. Sequence alignment of signal peptides from different alleles of PspC. Different signal peptide sequences were aligned using Clustal omega. The box shows the twin arginine motif present specifically in PspC2 or similar alleles.

CLUSTAL O(1.2.4) multiple sequence alignment

```

PspC6.13      MFASKNERKVHYSIRKFSIGVASVAVASLFMGSVVHA
PspC8.1       MFASKSERKVHYSIRKFSIGVASVAVASLFMGSVVHA
PspC8.2       MFASKSERKVHYSIRKFSIGVASVAVASLFMGSVVHA
PspC8.3       MFASKSERKVHYSIRKFSIGVASVAVASLFMGSVVHA
PspC8.4       MFASKSERKVHYSIRKFSIGVASVAVASLFMGSVVHA
PspC11.2      MFASKNERKVHYSIRKFSIGVASVAVASLFMGSVVHA
PspC11.3      MFASKNERKVHYSIRKFSIGVASVAVASLFMGSVVHA
PspC2.5       MFASKSERKVHYSIRKFSIGVASVAVASLFMGSVVHA
PspC3.1       MFASKSERKVHYSIRKFSIGVASVAVASLFMGSVVHA
PspC3.6       MFASKSERKVHYSIRKFSIGVASVAVASLFMGSVVHA
PspC3.7       MFASKNERKVHYSIRKFSIGVASVAVASLFMGSVVHA
PspC3.8       MFASKSERKVHYSIRKFSIGVASVAVASLFMGSVVHA
PspC3.9       MFASKNERKVHYSIRKFSIGVASVAVASLFMGSVVHA
PspC3.10      MFASKNERKVHYSIRKFSIGVASVAVASLFMGSVVHA
PspC3.11      MFASKSERKVHYSIRKFSIGVASVAVASLFMGSVVHA
PspC3.12      MFASKNERKVHYSIRKFSIGVASVAVASLFMGSVVHA
PspC6.14      MFASKSERKVHYSIRKFSVGVASVVASLFLGRVVHA
PspC6.4       MFASKSERKVHYSIRKFSVGVASVVASLFLGRVVHA
PspC5.2       MFASKSERKVHYSIRKFSIGVASVVASLFMGSVVHA
PspC5.1       MFASKSERKVHYSIRKFSIGVASVVASLFMGSVVHA
PspC3.13      MFASKSERKVHYSIRKFSIGVASVVASLFMGSVVHA
PspC3.5       MFASKKERVHYSIRKFSIGVASVVASLFMGSVVHA
PspC2.4       MFASKSERKVHYSIRKFSIGVASVVASLFMGSVVHA
PspC2.3       MFASKSERKVHYSIRKFSIGVASVVASLFMGSVVHA
PspC2.2       MFASKSERKVHYSIRKFSVGVASVVASLFMGSVVHA
PspC1.1       MFASKSERKVHYSIRKFSIGVASVAVASLFLGGVVHA
PspC6.11      MFASKSERKVHYSIRKFSIGVASVAVASLFLGGVVHA
PspC11.1      MFKSNHERRMRYRYSIRKFSVGVASVAVASLFMGSVVHA
PspC9.4       MFKSNHERRMRYRYSIRKFSVGVASVAVASLFMGSVVHA
PspC9.3       MFKSNHERRMRYRYSIRKFSVGVASVAVASLFMGSVVHA
PspC9.2       MFKSNHERRMRYRYSIRKFSVGVASVAVASLFMGSVVHA
PspC9.1       MFKSNHERRMRYRYSIRKFSVGVASVAVASLFMGSVVHA
PspC10.1      MFKSNHERRMRYRYSIRKFSVGVASVAVASLFMGSVVHA
PspC6.8       MFASKNERKVHYSIRKFSIGVASVAVASLFLGGVAHA
PspC6.1       MFASKSERKVHYSIRKFSIGVASVAVASLFLGGVAHA
PspC6.12      MFASKSERKVHYSIRKFSIGVASVVASLFLGGVVHA
PspC6.10      MFASKSERKVHYSIRKFSVGVASVVASLFLGGVVHA
PspC6.7       MFASKSERKVHYSIRKFSIGVASVVASLFLGGVVHA
PspC6.6       MFASKSERKVHYSIRKFSVGVASVVASLFLGGVVHA
PspC6.2       MFASKSERKVHYSIRKFSVGVASVVASLFLGGVVHA
PspC6.9       MFASKSERKVHYSIRKFSIGVASVVASLFLGGVVHA
PspC6.5       MFASKSERKVHYSIRKFSIGVASVVASLFLGGVVHA
PspC4.1       MFASKSERKVHYSIRKFSIGVASVVASLFLGGVVHA
PspC7.1       MFKSNYERKMCYSIRKFSIGVASVAVASLFMGSVVHA
PspC7.2       MFKSNYERKMCYSIRKFSIGVASVAVASLFMGSVVHA
PspC7.3       MFKSNYERKMCYSIRKFSIGVASVAVASLFMGSVVHA
PspC7.4       MFKSNYERKMCYSIRKFSIGVASVAVASLFMGSVVHA
** *  **  *  *  *  *  *  *  *  *  *  *  *  *  *  *  *

```

Suppl. Fig S3. Graphical illustration of PspC1 and PspC2.

PspC1 contains one RICH domain while PspC2 contains two RICH domains (Pfam-PF05062) in its N-terminal alpha helical part. Abbreviations: PRR-proline rich repeats, SP-signal peptide, CBD-choline binding domain, LPxTG-LPxTG anchor domain.

Suppl. Fig S4. Full protein sequences of PspC1 and PspC2 in strain BHN418.

BHN418 PspC1:

```
MFASKSERKVHYSIRKFSIGVASVVVASLFLGGVVHAEVGGRRNTPVTSSGQDISKKYADEVESHLKILSEIQTQLDRKRHTKTV
ALINELQDIKKTLYLNVLNVLKEKSELPSKIKAKLEVAFDQFKKDTLKPGEKVAEAEKKVAEAKKKAEDQKEEDRRNYPTNTYKTLLEL
EIAESDVKVKAEAELELVNEEAKPGNEEKIKKAKAKVESEKAEAIRLEEIKTDREEAKRKADAKLKEAVENNAATSEQGEPKRRVTKRG
VLGEPATPDKKENDAKSSDSSVGEETLPSPLKPEKKVAEAEKKAKDQKEEDRRNYPTNTYKTLLEIAESDVKVKAEAELELVKEEA
KESRNEEKVKQAKAKVESKKAETRLEKIKTDRKKAEEAKRKAEEEDKVKEKPAEQQPAPAPQPEKPAPKPEKPAPAPKPENPA
EQPKAEKPADQQAEEYARRSEEEYNRLTQQQPPKTEKPAQPSTPKTGWKQENGMWYFYNTDGS MATGWLQNNGSWYYL
NSNGAMATGWLQNNGSWYYLNANGSMATGWLQNNGSWYYLNANGSMATGWLQNNGSWYYLNANGSMATGWLQNN
GSWYYLNANGSMATGWLQYNGSWYYLNANGSMATGWLQYNGSWYYLNANGSMATGWLQYNGSWYYLNANGSMATG
VWKDGD TWYYLEASGAMKASQWFKVSDKWYYVNGSGALAVNTTVDSYRVNANG EWVN
```

BHN 418 PspC2:

```
MFKSNHERRMRYSIRKFSVGVASVAVASLFGMSVVHATEKEGSTQAATSFNRGNGSQAEQREGELDLERDKAMKAVSEYVVGK
MVRDAYVKS DRKRHKNTVALVNQLGNIKNRYLNEIVHSTSKSQLQELMMKSQSEVDEAVSKFEKDSFSSSSSGSSTKPEQPQE
NPEHQKPTTSPDTPSPQPEGKPSVPDINQEKEKAKLAVVTYMSKILDDIQKHHLQKEKHRQIVALIKELDELKQALSEIDNV
NTKVEIENTVHKIFADMDAVVTFKFKGLTQDTPKEPGNKKPSAPKPGMQPSPQPEVKPQLEPKPEVKPQPEKPKPEVKPQPE
KPKPEVKPQPEKPKPEVKPQPEKPKPEVKPQPEKPKPEVKPQPEKPKPEVKPQPEKPKPEVKPQPEKPKPEVKPQPEKPKPE
QPEKPKPEVKPQPEKPKPEVKPQPEKPKPEVKPQPEKPKPEVKPQPEKPKPEVKPQPEKPKPEVKPQPEKPKPEVKPQPEKPKPE
VKPQPEKPKPEVKPQPEKPKPDNSKPQADDKKPSTTNLSKDKQPSNQASTNEKATNKPKKSLPSTGSISNLALEIAGLLLAGA
TILAKKRMK
```

Suppl. Fig S6. Identification of the FH binding domain of PspC1 and PspC2.

(a) An array of 48 overlapping peptides was synthesized and covalently spotted onto a cellulose membrane covering the N-terminal domain of PspC1 and PspC2. Each spot contained a 15 amino acids long peptide with three overlapping amino acids to the next spot. Spots showing more than 80 percent binding were considered as full binders. Sequences of amino acids from the binding peptides are shown with binding motif represented by an empty box.

(b) Structure based alignment of N terminal domains of different PspCs. The N-terminal sequence of two copies of PspC present in clinical isolates showing high FH binding level (PspC1 and PspC2) and other members of the PspC family (PspC from TIGR4, SpsA_type1^{S1}, D39 and HIC^{S2}) were aligned with the structure based alignment tool PROMALS3D^{S3}. The recently described three dimensional structure of the FH binding part of TIGR4 PspC was used as a template for the structure based alignment. All 17 residues of PspC in TIGR4 shown to interact with human FH are shaded in black. Shaded in grey are the residues that are conserved in other PspCs. Key residue Tyrosine⁹⁰ described in the crystal structure is indicated by a star. Family A and family B of PspCs are labeled accordingly with a line separating both families. Empty boxes shows conserved residues of FH binding motif found by peptide array analysis on PspC1 and PspC2. Consensus amino acids and consensus secondary structures are mentioned in the last rows.

a

b

PROMALS3D alignment (Sequences in aligned order)

PspC_Tigr4	1	MFASKSERKVHYSIRKFSVGVASVVVASLVMGVSVVHATENEGATQVPTSSNRANESQAEQGEQPKKLDSEKDKAR	75
SpsA_type1	1	MFASKSERKVHYSIRKFSIGVASVVVASLVMGVSVVHATENEGSTQAATFSNMANKSQTEQGE---INIERDKAK	71
PspC2_BHN191/BHN418	1	MFKSNHERMRYSIRKFSVGVASVAVASLVMGVSVVHATEKEGSTQAATSFNRGNGSQAEQGE---ELDLERDKAM	72
HIC_A66	1	MFASKNERKVHYSIRKFSIGVASVAVASLVMGVSVVHATEKEVTTQVATSSNKANKSQTEHMK-----AA	64
PspC_D39	1	MFASKSERKVHYSIRKFSIGVASVAVASLVMGVSVVHATENEGSTQAATSSNMAKTEHRKA-----AK	62
PspC1_BHN191	1	MFASKSERKVHYSIRKFSIGVASVVVASLFLGGVVHAEVGGRRNTPTVTSSGQDISKKYA-----	60
PspC1_BHN418	1	MFASKSERKVHYSIRKFSIGVASVVVASLFLGGVVHAEVGGRRNTPTVTSSGQDISKKYA-----	60
Consensus aa:		MFASKsERKVHYSIRKFSIGVASVhVASLhghgTVVHApE...GpspsghoSs..s.Sp.b.....	
Consensus ss:		hhhhhhhhhhhhhhhhhhhhhhhhhhhhhh eeee hhh hhhhhhh	
			
PspC_Tigr4	76	KEVEEYVKKILGES*YAKSTKSTKTTWVALNPLNNENKLYLNKIVESTSESQ----LQILMMESRSKVDEAVSK	145
SpsA_type1	72	TAVSEYKEKKVSEIYTKLEDRHKITVDLANKLQEIKNFYLNKIVQSTSKTE----IQGLITPSTRSKLDEAVSK	141
PspC2_BHN191/BHN418	73	KAVSEYVGMVRDAYVKSDRKRHKNTVALNQLGNIKRNLNEIVHSTSKSQ----LQELMMKSQSEVDEAVSK	142
HIC_A66	65	KQVDEYIEKMLSET--QLDRRKHTQNVGLLTKLGAIKTEYLRGLSVSKEKS----TALPSEIKEKLTAAFEQ	131
PspC_D39	63	QVVDEYIEKMLREI--QLDRRKHTQNVGLNKLKSAIKTKYLRLELVLEKS----KDELPEIKAKLDAAFEK	129
PspC1_BHN191	61	DEVKSHLEEMFGI--QLDRRKHTQNFNLKLSKIQTKYLRLELVLSKSELTLETKEELLSKTKAELDAAFEQ	133
PspC1_BHN418	61	DEVESHKKILSEBIQTQLDRKRHTKTVALINELQDIKTKTYLYNLVLEKS----ELPSKIKAKLEVAFDQ	127
Consensus aa:		p.Vpp@LcKkHL...p.s+++Hp.sVSLI.cL.IKscYlppLs.ppsS...bpbL.c.+tcL-.Ahpp	
Consensus ss:		hh hhh hhhhhhhhhhhhhhhhhhh	
			
PspC_Tigr4	146	FEKDSSTSSSSSSSTKPK-----EASDTAKP-----NKPTEPG-----EKV-----	180
SpsA_type1	142	YKAPSSSSSSSSSTKPK-----EASDTAKP-----NKPTELE-----KKVAEAE-----	180
PspC2_BHN191/BHN418	143	FEKDSFSSSSSSSTKPK-----ETPOFENPE-----HOKPTTPS-----PDTK-----	180
HIC_A66	132	FHKDTLKS--G--RVAE-----AQKKV--KDQKEAKQTEALIVKHKGR-----ETDLDLRKKAK--	180
PspC_D39	130	FHKDTLKP--G--EHVAEAKKVVAEAKKKVEDQKEEDR-----RNYPTNTYKTLELEI-----AE--	180
PspC1_BHN191	134	FHKDTLKL--G--EHVAEAEKVVAEAKKKVKAQKEEDH-----RNYPTNTYKTLELEI-----AE--	180
PspC1_BHN418	128	FHKDTLKP--G--EHVAEAEKVVAEAKKKVEDQKEEDR-----RNYPTNTYKTLELEI-----AESD	180
Consensus aa:		fKdD.ps...G...pkS...spp.ps.....	
Consensus ss:		hhh hhhh hhhhhhhhhhhhhhhhhhh	

Have the authors considered that factor *h* is also discussed to interact and bind to CR3 adhesion receptor ?

Response: We studied whether pure FH and pure C3b would affect the uptake into THP-1 macrophages of wild type BHN418 or its isogenic mutants in PspC1 and PspC2, but found no significant differences in the uptake between the strains (new **Suppl. Fig. S14**). We observed though a lower uptake when the bacteria were incubated with pure C3b than with normal human serum which might be due to that incubation with pure C3b offers much less C3b in comparison to when serum is used, leading to lower opsonization. We also used A549 epithelial cells and found that pure FH did not affect adhesion to these cells, see figure below.

Suppl. Fig. S14. Pneumococcal uptake by THP-1 cells in the presence of pure FH and pure C3b. Uptake by THP-1 derived macrophages of pneumococcal strain BHN418 or its isogenic *pspC* deletion mutants in the presence of (a) purified FH or (b) purified C3b. No significant difference was found between the strains.

Figure. Adhesion of pneumococcal strains to A549 epithelial cells with or without addition of pure FH.

Minor

Introduction. C3 is not a complement factor

Response: In agreement with the reviewer comment, this has been changed accordingly.

The specificity of the new antisera generated should be shown

Response: To address the reviewer's concern we have included a supplementary figure with the western blot of the whole cell lysate of BHN418 with the respective anti-PspC1 and anti-PspC2 antibodies (new **Suppl. Fig. S13**)

Suppl. Fig. S13. Detection of the degradation product C3d.

Western blot analysis showing deposited C3d in wt strain BHN418 and in its isogenic *pspC* mutants, BHN418 Δ *pspC1*, BHN418 Δ *pspC2* and BHN418 Δ *pspC1* Δ *pspC2*.

For Supplementary Table S2: the Bicaore experiments the real binding data and the langmuir binding models would be informative. Thus the fitting parameters of the experiment should be included and the concentration range tested.

Response: In agreement with the reviewer's comment, we have included a supplementary figure with the details of the BIACORE experiment (new **Suppl. Fig. S5**).

Suppl. Fig. S5. Surface plasmon resonance/BIACORE analysis of PspC-FH interactions.

Representative sensograms of surface plasmon resonance analysis/BIACORE analysis of PspC-FH interactions for three different PspC alleles (PspC1, and PspC2 in strain BHN418, and PspC from D39). Response (resonance unit-RU) versus time plot was obtained by injecting five increasing concentrations of recombinant PspC proteins. Colored curves were obtained by fitting data with a 1:1 binding model. See Methods for a more detailed description.

PspC1 BHN418

Curve	ka (1/Ms)	kd (1/s)	KD (M)	Rmax (RU)	Conc (M)	tc	Flow (ul/min)	kt (RU/Ms)	RI (RU)	Chi² (RU²)	U-value
Cycle: 6	6,854E+5	6,221E-5	9,076E-11	65,55	9,375E-10	1,726E+9	30,00	5,363E+9	-0,03424	0,126	1
					1,875E-9				-0,3875		
					3,750E-9				-0,4645		
					7,500E-9				-0,7131		
					1,500E-8				-0,6823		

PspC2 BHN418

Curve	ka (1/Ms)	kd (1/s)	KD (M)	Rmax (RU)	Conc (M)	tc	Flow (ul/min)	kt (RU/Ms)	RI (RU)	Chi² (RU²)	U-value
Cycle: 4	4,11E+06	5,47E-05	1,33E-11	37,91	6,25E-10	6,30E+13	30	1,96E+14	0,2367	0,455	2
					1,25E-09				-0,1538		
					2,50E-09				-0,5736		
					5,00E-09				-0,2998		
					1,00E-08				1,387		

PspC D39

Curve	ka (1/Ms)	kd (1/s)	KD (M)	Rmax (RU)	Conc (M)	tc	Flow (ul/min)	kt (RU/Ms)	RI (RU)	Chi² (RU²)	U-value
Cycle: 6	4,836E+5	4,281E-5	8,854E-11	43,03	1,600E-9	1,752E+21	30,00	5,443E+21	-0,1647	0,754	2
					4,810E-9				-0,2538		
					1,440E-8				0,02706		
					4,330E-8				-2,374		
					1,300E-7				-1,903		

For Supplementary Figure S2: in panel A what peptide spot represents PspC1 and which PspC2 +

Response: We have now labelled what peptide spot represents PspC1 and which PspC2 in the respective panels in Suppl. **Fig S6a**, see the figure in the response above.

Be careful with a generalization of linear peptide binding motifs as binding residues. Please be consistent in using C5b-9 vs MAC (e.g. page 10, line 290)

Response: We have changed the text accordingly as suggested by the reviewer.

Reviewer#2

Here, the authors use STED microscopy to study the spatial distribution of C3 (and related proteins) in Bacteria. My expertise is not in the biology so I can't comment with authority on the novelty, but I havent seen similar studies and so it seems a nice advance.

We thank the reviewer for this comment.

On the imaging side, while the microscopy itself seems quite well done (and quite well described), there is a worrying lack of quantification throughout (other than some basic cell counting). If this were done (and of course, if the results hold up) I would consider this manuscript acceptable for publication. The corrections necessary are quite extensive however.

Most notably is a lack of rigorous co-localisation analysis. This require the calculation of the Pearson's or Mander's colocalisation coefficients for each image and then performing statistical t-tests to test for significant differences between those coefficients in different conditions. Additionally - the authors describe sub-cellular localisation quite informally. It should be possible to quantify how "even" a staining pattern is and (again) calculate significances between conditions. This might involve the use of additional stains to highlight specific cellular organelles/regions and then test for degrees of colocalisation. Statements about cell shape and morphology can also be quantified.

Response: In agreement with the concerns by this reviewer we have made new analyses calculating differences for co-localization patterns for experiments shown in Figure 4 and 5. New supplementary figures have been included showing the co-localization analysis of different co-staining experiments (**Suppl. Fig. S11 and S12**). The co-localization calculations were carried out using custom written code in MATLAB R2013b. The calculations, analysis and thresholding of images followed the outline described in detail in reference 36. The co-localization coefficients were based on Image Cross-Correlation Spectroscopy (ICCS) (reference 37) as well as Pearsons correlation (reference 38). By ICCS, co-localization coefficients are obtained for both the red and the green channel. These coefficients are more suited for co-localization analysis when there is an excess of one of the fluorescent species compared to the other, i.e if there is more green labeled molecules than red labeled molecules

in the images, or vice versa. In the recorded STED images in this study, there was almost always more red labeled species than green labeled species, and we therefore used the ICCS_{GREEN} coefficient as a measure for co-localization. The ICCS_{GREEN} coefficient takes on values between 0 and 1, where 0 is no co-localization and 1 is perfect co-localization between the channels. Images were processed by applying a smoothing Gaussian filter in order to reduce noise before the co-localization analysis.

For statistical significance we used a two sided student t-test to test the null-hypotheses that the co-localization coefficients from two different samples have the same mean. This test yields a p-value giving the probability to for the null-hypothesis being true. If the p-value < 0.05 the null-hypothesis is rejected with a *significance, with a p-value<0.01 the null-hypothesis is rejected with a **significance, and if the p-value<0.001 the null-hypothesis is rejected with a ***significance.

Suppl. Fig. 11. Calculation of co-localization between PspC1 and PspC2 and FH.

The co-localization coefficient for the staining of PspC1 and PspC2 and FH, as shown in Figure 4, was calculated. PspC2 and FH showed a low co-localization coefficient in wt BHN418, while a significantly higher co-localization coefficient was found in the mutant lacking PspC1, BHN418Δ*pspC1*. FH and PspC1 in wt BHN showed a high co-localization coefficient. Number of cells included in the analyses: PspC2: BHN418 (n=106), and BHN418Δ*pspC1* (n=95). PspC1: BHN418 (n=111).

Suppl. Fig. 12. Calculation of co-localization between the capsule and C3b.

The co-localization coefficient was calculated for the capsule and C3b for images shown in Figure 5. The capsule and C3b showed a low co-localization coefficient for wt BHN418 and its isogenic *pspC* mutants. Number of cells included in the analyses: BHN418 (n=32), BHN418Δ*pspC1* (n=57), BHN418Δ*pspC2* (n=28), BHN418Δ*pspC1*Δ*pspC2* (n=35). No significant differences in co-localization were found according to the t-test.

The final important one is the staining intensities - describes as "low" for example. These should be quantified and (since the staining intensity is fairly arbitrary) probably be given as fold changes (with significances) between conditions.

Response: We have used FACS to quantify the relative C3b deposition and FH binding. It is difficult to quantify staining intensities on the bacteria in absolute terms from the images. Hence in agreement with the reviewer comment we have now changed the text and do not use 'low intensity'.

Finally, the authors should be careful with Figure formatting e.g. some axes are small and quite hard to read e.g. 5b,d.

Response: According to the suggestions by the reviewer we have changed the axis in the **Figures 5b, 5d, 7e and 7f**, to make it easier for the readers.

The M&M says a Leica microscope with 100X resolution was used. this is the magnification not the resolution.

Response: The reviewer is correct and we have changed this accordingly in the Methods section.

Reviewer#3

The study aims to elucidate if the immune defence mechanisms exerted by the capsular polysaccharide (CPS) of pneumococci and the Factor H binding proteins (FHBPs), are spatially coordinated on the bacterial cell surface. The CPS protects against phagocytosis and FHBPs prevent binding of C3b deposition, which otherwise would activate the complement cascade. Here the authors have analysed deposition of C3b and Factor H binding in wild type pneumococci, stains with capsule, and in capsule lacking pneumococci. The study is unique and very interesting because host protein

binding was visualized by CLSM and STED, supported by classical binding studies. In addition, two types of PspC proteins, named PspC1 and PspC2, are produced by the selected strain BHN418 (a serotype 6B) and importantly, the mechanism of cell wall anchoring is different for both. While PspC1 is a choline binding protein, PspC2 is anchored by the sortase. The study now figured out when and where PspC1 or PspC2 are located at the bacterial cell surface (in the wild type vs unencapsulated strain) to be accessible for the interaction with Factor H (thereby avoiding C3b deposition). The results demonstrate nicely 1. that the distribution of PspC1 and PspC2 on the surface of wild type pneumococci is different and 2. that PspC1 is located at the division sites, while PspC2 is located at the poles or distributed among the cell surface, similar to RrgB (pilus-1 backbone). These findings are quite surprising and new, because none of the former studies has illustrated the distribution of PspC molecules (or other choline-binding proteins) on the bacterial cell surface. This PspC localization in CPS containing bacteria has consequences for the interaction with Factor H and deposition of C3b as well. C3b and Factor H binding was detected in regions lacking CPS, hence in the septa of dividing cells, where only PspC1 is located. Thus, the TCC (C5b-C9) was also only detected in the division septa of a few bacteria. In consequence the findings show that the mechanism of immune evasion by the CPS and PspC1 or PspC2 is a coordinated process, and that PspC1 is probably masked by the CPS but exposed at the division sites to prevent attack by C3b and the MAC. In general the study is highly interesting and provides new insights into the mechanistic functions of a unique class of surface protein. The distribution and exerted function(s) strongly depend on the anchoring motif. However, the study lacks several details regarding the applied methods and host cells used, and the reviewer has also some concerns about the specificity of tools used and methods applied to elucidate the PspC functions.

We thank the reviewer for stating that our study is highly interesting and provides new insights into the mechanistic functions of a unique class of surface protein.

1. The summary is not very informative because it contains mostly state of the art knowledge. The breaking news are not mentioned clearly and this makes it difficult to understand the importance of the results

Response: In agreement with the reviewer, we have changed the summary accordingly to be more informative on our results.

2. In general the titles in the results section are not well defined. More specific titles indicating the major findings have to be given

Response: As suggested by the reviewer, headlines of the result section have been changed to be more specific.

3. C3b deposition: the TEM in Figure 1 shows dark regions, probably of C3b deposition. Here immunogold and appropriate controls are the better choice

Response: As suggested by the reviewer we have now performed immunogold staining of C3b in wild type BHN418 and of non-encapsulated BHN418 after incubation with 20% NHS and TEM (new **Suppl. Fig. S1**, see above in response to Reviewer 1)

4. How can C3b be deposited on the cell surface and detected by antibodies, when Factor H is also bound and inactivates C3b into C3bi?

Response: Our data show that considerably less C3b is deposited on the cell surface of bacteria expressing PspC that bind FH, than on bacteria not expressing PspC (**Figure 5**). When FH is bound it is not for sure that all FH reaches all C3b and we do not expect that all C3b should be inactivated. This is less likely for spatial and temporal reasons.

5. The authors used polyclonal antibodies raised against PspC1 and PspC2 and used these antibodies to differentiate between PspC1 and PspC2. According to the document and the partial sequence provided by PspC types it is not clear at all why there is a specific binding of the antibodies to PspC1 or PspC2. An additional experimental evidence has to be given to demonstrate protein specificity.

Response: As suggested by the reviewer we have now included a supplementary figure with the western blots of whole cell lysate of BHN 418 with anti-PspC1 antibody as well as anti-PspC2 antibody showing the specificity of the antibodies for respective proteins (new **Suppl. Fig. S7**).

Suppl. Fig. S7. Specificity of anti-PspC1 and anti-PspC2 antibodies.

Western blot analysis showing the specificity of the anti PspC1 and anti-PspC2 antibodies used in the study when probed against cell lysate of wt BHN418. Calculated molecular weight of PspC1 and PspC2 is 80.93 kDa and 66.92 kDa respectively.

Line 146 mentioned that PspC2 belongs to the TIGR4 PspC, which has (according to the performed sequence alignment done by the reviewer) a high homology to the D39 PspC (these are orthologues!)

Response: When we did sequence alignments to define two families of PspC, we checked only the sequence homology of the FH binding domains of different PspCs (**Suppl. Fig 6**, see above in response to Reviewer 1). PspC2 belongs to the group that lacks a crucial tyrosine residue shown to be involved in its interaction with FH (shown by asterisk in **Suppl. Fig S6b**). Hence, we grouped them into Family A and B.

6. In line with this, the total sequences of PspC1 and PspC2 of BHN418 has to be provided in the supplement as well as a sequence alignment of the N-terminal regions

Response: In agreement with the reviewer we have included a supplementary figure with the full protein sequences of BHN418 PspC1 and PspC2 (new **Suppl. Fig. S4**, see response to Reviewer 1 above). There is also a sequence alignment of the N-terminal domains of these proteins in **Suppl. Fig 6b**.

7. PspC proteins have a modular organization with different function domains – more information a needed for non-experts on pneumococci and PspC to understand the molecules and the findings by Pathak et al

Response: To address this comment by the reviewer, we have included a supplementary figure (**Suppl. Fig. S3**, see response to Reviewer 1 above) showing the modular organization of PspC1 and PspC2.

8. Fig. 4: the authors constructed pspC mutants and showed binding of Factor H only to pneumococci expressing PspC1 in the histograms. However, the immunofluorescence images show binding also to PspC2. Please explain this discrepancy –

Response: We show in **Figure 4a** that PspC1 is the major FH binding allele in BHN418. However, there is still a little FH binding left in the PspC1 deletion mutant (**Fig 4a**, second panel), not found in the double mutant (**Fig. 4a, 4th panel**), suggesting that PspC2 is a binder in the absence of PspC1.

9. A major question for PspC1 and PspC2 remains the killing of pneumococci by formation of the MAC in wild type and mutant pneumococci (bactericidal assay)

Response: We studied the bactericidal activity after incubation of bacteria with 20% NHS, but observed no detectable killing. Bactericidal activity was detected by live dead stain (green-live, red-dead) along with detection of C5b-9(MAC) (blue)(Figure below). We found some dead bacteria that co-stained with C5b-9 (white arrow), but we could also detect live bacteria with C5b-9 detection (red arrows). Therefore our data remains inconclusive.

10. Opsonophagocytosis assays: the results of the opsonophagocytosis assay are not clear. As a result of serum incubation pneumococci bind several components of the serum. In wild type CPS expressing bacteria they probably bind Factor H and other components. Factor H is known to recognize integrins on phagocytes and thus, enhanced uptake by e.g. PMNs has been described.

Response: In **Figure 6a** we study uptake of bacteria by THP-1 macrophages. We observe that the mutant in PspC1 is defective in binding to FH, has a higher uptake, suggesting that FH does not play a major role in the uptake. Also, we studied uptake of wild type and mutant bacteria by THP-1 macrophages in the presence of pure FH or pure C3b and we found no differences in the uptake (new **Suppl. Fig. S14**, see response to Reviewer 1 above).

In the view of the reviewer the effects of Factor H and the other serum components has not been considered. How is phagocytosis affected in the presence of C3b alone or Factor H alone?

Response: As suggested by the reviewer we performed uptake assays using THP-1 macrophages in the presence of pure FH or pure C3b. We observed no significant differences in bacterial uptake between wild type BHN418 and the different PspC mutants when bacteria were incubated with either pure FH or pure C3b (new **Suppl. Fig. S14**, see response to Reviewer 1 above).

11. When the reviewer is correct, both PspC molecules possess the binding domain for the polymeric Ig receptor-no. However, this receptor is not expressed by A549. Therefore, the results of this paragraph are unclear to the reviewer, because neither the PspC-pIgR interaction nor the Factor H mediated adherence was investigated. Alternatively, why is there a dramatic effect on adherence when the appropriate receptor for PspC is missing?

Response: In response to this question we studied adhesion of BHN418 and its *pspC* mutants to nasopharyngeal Detroit cells, which are known to present the pIgR receptor (new **Suppl. Fig. S20**). The adhesion assay showed similar adhesion patterns as that of A549 cells where we show that PspC2 is important for bacterial adhesion. Deletion of PspC1 possessing the pIgR binding motif had no significant effect on the bacterial adhesion pattern.

Suppl. Fig. S20. Pneumococcal adhesion to Detroit nasopharyngeal epithelial cells

Adherence of wild type BHN418, or isogenic *pspC* mutants, to Detroit 562 cells. Graph shows mean \pm SEM of four independent experiments. ***, $p < 0.001$.

12. Figure 7 lacks the immunoblot for PspC2

Response: As suggested by the referee the blot for PspC2 is now included in **Figure 7**.

Fig. 7d. Western blot analysis showing the expression level of PspC1 and PspC2 in wt BHN418 and in the signal peptide switch mutant, BHN418 Δ sp2pspC1. GAPDH was used as a loading control.

13. Page 12: the first part of the discussion is a repetition of the introduction. These parts can be combined

Response: In agreement with the reviewer comment this has been changed accordingly

Minor suggestions

- *Not all choline binding proteins contain the choline binding motifs at the carboxy-terminal part*

Response: In agreement with this comment we have now changed this sentence in the introduction.

- *The study lacks important citations and findings of original studies. PspC is, according to the literature, a multifunctional protein and divided into different subclasses. In addition to bind Factor H, here with prototypes of the “conventional” PspC and the sortase-anchored Hic, both types have been shown to bind vitronectin, which is also a complement inhibitor. Binding studies by SPR even analysed in depth the protein-protein interactions*
- *The binding regions (not the motifs) have been described for PspC and Hic. The simultaneous binding or even the competitions between Factor H and vitronectin binding or sIgA binding have also been reported. These facts are important to understand the function of PspC/Hic in invasive infections*
- *In addition, PspC (but not the Hic subtypes) link *S. pneumoniae* with the pIgR (several publications). The consequences of this interaction or with free sIgA has been reported*
- *The above mentioned findings are not referenced in the Introduction, but represent an important aspect when incubating pneumococci with serum.*
- *It is therefore not possible to restrict the analysis to Factor H binding when incubating pneumococci with serum*

Response: To address this comment by the reviewer we have now inserted new references including studies of vitronectin, sIgA and pIgR interactions with PspC in the introduction.

- *The sensorgrams of the SPR studies have to be shown and the evaluation of the data has to be provided. All details of the SPR experiments are missing (coupling procedure, analyte concentration and flow...) –*

Response: In agreement with the reviewer comment we have included details of the SPR experiments in a new supplementary figure (new **Suppl. Fig S5**, see above in response to Reviewer 1).

- *Line 330: correct the sentence “We show here show...”*

Response: We thank the reviewer for this comment. This has been changed accordingly.

- *Line 342: PspC is not decorating choline residues on teichoic acids*

Response: This has been changed in the modified manuscript.

• *In general: check the references and citations*

Response: References and citations have now been checked by the authors in the modified manuscript.

• *Are pneumococci killed by using only gentamicin?-*

Response: Gentamicin is reported by others to kill pneumococci (*Djurkovic S et al Antimicrobial Agents and Chemotherapy. 2005;49(3):1225-1228*). Additionally, we tested whether gentamicin treatment kills pneumococci in our assay, and we could not detect any CFUs in the supernatant of cells, after gentamicin treatment in our experiments (data not shown).

Reviewers' comments:

Reviewer #1 (Remarks to the Author):

The authors have addressed the issue raised and have adequately responded to all points. In my opinion the manuscript is almost ready to be acceptable for publication in Nature Communication.

One issue remains as summarized in Figure 8. I do understand from the data presented in the manuscript that for intact BHN418 pneumococci only PspC1, but not PspC2 is exposed and accessible in the intact bacterium and PspC1 binds Factor H. Meaning that PspC1 is accessible at the division septum and PspC2 is masked. One way as presented in the upper part of the figure for single cocci is masking by the capsule. The pspC1 knock out (lower panel, upper right side) PspC2 located at the pole now binds Factor H. In this case PspC2 should be accessible to plasma and to the outside world. In the graph however PspC2 is still masked by the capsule. Can this issue be addressed and solved.

Reviewer #2 (Remarks to the Author):

The authors have addressed some of my concerns, but unfortunately some still remain and have not been addressed, and therefore at the moment, I cannot recommend publication. While the authors now perform quantitative colocalization analysis, there is still a lack of quantification and an informal "this looks like" style to their observations. For example:

"By comparing STED-images of C3b and C5b-9 deposition there seemed to be a larger distance between the capsule and the edges of C5b-9 bands than between the capsule and C3b (Fig.2c,d)."

What does "seemed to be" mean here? Statistically - is there, or isn't there?

Here are a couple more examples:

"The distance between a PspC1 band and its older pole was the same, about 0.5 μm , irrespective of cell length"

Is it the same? by what statistical test? And instead of "about 0.5", they could give the actual value with S.D.

"Importantly, a much larger area underneath the capsular layer contained deposited C3b when compared to the wt."

But *how much larger*? is that increase statistically significant? As it reads it just sounds that it "looks larger" in a few images, and the authors have determined informally that it is a real change, not the result of statistical fluctuation.

There are many more examples.

Again, I'm not saying any of these conclusions are not true, but the statements are not backed up by quantification and statistical tests. If you say "the area of protein X is larger in sample A than sample B" that statement *has* to be accompanied by quantification of *how much larger*, in what n-numbers and has to have a statistical test of significance.

Until this is done, I'm afraid I still cannot recommend publication.

Reviewer #3 (Remarks to the Author):

The authors have done a great job to respond adequately to the reviewers' comments. They have performed additional experiments and now indicate by providing additional data the spatial distribution of the different PspC variants in the presence of the CPS.

I have only two but minor comments:

1. the PspC models in the supplement (Fig. S3) can probably be modified in a way that not only the RICH are shown (which contain the FH binding motifs) but also binding region(s) of the polymeric Ig receptor. Studying the sequences the PspC1 contains two R domains and the RICH precedes these domains. The PspC2 variant contains two RICH domains – which seem to more or less directly after the signal peptide and N-terminal to the PRR, however, there is no R domain. This information should be included in the models and legends.

2. the sensorgrams (not sensograms) of the Biacore data are raw data. The data have to be processed and the kinetic data should be shown.

Point-by-point response to the comments made by the reviewers

Reviewer #1

The authors have addressed the issue raised and have adequately responded to all points. In my opinion the manuscript is almost ready to be acceptable for publication in Nature Communication.

We thank the reviewer for this positive comment stating that the manuscript is almost ready to be acceptable for publication in Nature Communication.

One issue remains as summarized in Figure 8. I do understand from the data presented in the manuscript that for intact BHN418 pneumococci only PspC1, but not PspC2 is exposed and accessible in the intact bacterium and PspC1 binds Factor H. Meaning that PspC1 is accessible at the division septum and PspC2 is masked. One way as presented in the upper part of the figure for single cocci is masking by the capsule. The pspC1 knock out (lower panel, upper right side) PspC2 located at the pole now binds Factor H. In this case Pspc2 should be accessible to plasma and to the outside world. In the graph however Pspc2 is still masked by the capsule. Can this issue be addressed and solved.

Response: We thank the reviewer for this comment. In figure 8 our data suggest that the FH binding region of PspC2 is not exposed beyond the capsular layer. When PspC1 is lacking then FH enters at the septum and spread underneath the capsule to reach the pool of PspC2, which will not happen in the wild type strain since in this case FH will first reach and bind PspC1 at the septum. To clarify our hypothesis we have in figure 8 changed so that we have marked the WT with a line below in the figure to the left.

Reviewer #2

The authors have addressed some of my concerns, but unfortunately some still remain and have not been addressed, and therefore at the moment, I cannot recommend publication. While the authors now perform quantitative colocalisation analysis, there is still a lack of quantification and an informal "this looks like" style to their observations. For example:

"By comparing STED-images of C3b and C5b-9 deposition there seemed to be a larger distance between the capsule and the edges of C5b-9 bands than between the capsule and C3b (Fig.2c,d)." What does "seemed to be" mean here? Statistically - is there, or isn't there?

Response: To address this concern and comment we have now made additional analyses and calculations, see below. However, it is worth also to remember that many of the quantifications are made using FACS analyses and not only through images, please see below.

1. We have now measured the distance between the capsular signal and the C3 or C5b-9 signal in wild type BHN418 (figure 2c and 2d) as suggested by the reviewer. The difference in the distance was significant, as shown now in new Supplementary Figure S2, see below.

Suppl. Fig. S2. Calculations of the distance between the capsule, and C3b or C5b-9, respectively. The distances between the capsule and C3b (shown in figure 2c) or C5b-9 (shown in figure 2d) were calculated by manually drawing a line, in the image, perpendicular to the capsule and the complement to create a line profile. The distance was then estimated as the distance between the peaks in the line profile of the capsule and complement respectively. The distance between the capsule and C5b-9 (n=37) showed a significantly larger distance in comparison to the capsule and C3b (n=53).

"The distance between a PspC1 band and its older pole was the same, about 0.5 μ m, irrespective of cell length" Is it the same? by what statistical test? And instead of "about 0.5", they could give the actual value with S.D.

Response: In response to this question we have now, as suggested by the reviewer, inserted the actual value in the text. We found that the distance was 0.5013 +/-0.025. See line 177 on page 7 where we state 0.50±0.02.

"Importantly, a much larger area underneath the capsular layer contained deposited C3b when compared to the wt." But *how much larger*? is that increase statistically significant? As it reads it just sounds that it "looks larger" in a few images, and the authors have determined informally that it is a real change, not the result of statistical fluctuation.

Response: To address this concern by the reviewer, we have calculated the bacterial area covered by complement C3b in our microscopy images in Figure 5a, and inserted the data in new Suppl. Fig S13, see below. Importantly, we have also quantified complement binding for the strains using flow cytometry analysis FACS (figure 5b for figure 5a) and they are in agreement with the new data. Similarly, we did FACS analyses to quantify binding to support image data for other figures and these data all support our observations found in the microscopy images, shown in figures 4a, 5d, 7e and 7f.

Suppl. Fig. S13. Calculations of the bacterial surface covered by C3b. The area covered by C3b was calculated (see Methods for a more detailed description) for the images shown in Figure 5a. Wild type BHN418 (n=34) and BHN418 Δ pspC2 (n=33) showed significantly lower areas covered by C3b in comparison to the mutants BHN418 Δ pspC1 (n=48) and BHN418 Δ pspC1 Δ pspC2 (n=32).

There are many more examples.

*Again, I'm not saying any of these conclusions are not true, but the statements are not backed up by quantification and statistical tests. If you say "the area of protein X is larger in sample A than sample B" that statement *has* to be accompanied by quantification of *how much larger*, in what n-numbers and has to have a statistical test of significance.*

Response: To address this concern we have gone through the manuscript to see if there are additional calculations needed. As stated above many of the images are supported by FACS analyses. In Figure 2, we have made calculations of the distance between the capsule and C3b or C5b-9, now found in Suppl. Fig S2. In Figure 4, we have performed co-localization analyses between PspC1 and PspC2 and FH, now found in Suppl. Fig. 11. FACS analyses supports the data, found in Figure 4a. In Figure 5, we have done co-localization analyses between the capsule and C3b, now found in Suppl. Fig. 12 and calculated the bacterial surface covered by C3b, now found in Suppl. Fig. S13. FACS analyses supports the data, found in Figures 5b and d. Furthermore, we show FACS analyses data in figure 7e and 7f, supporting our conclusions.

We found three more items that we have strengthened by inserting more specific data. On page 5, line 102, we now state: ...some but not all division septa (18 out of 80 visible septa),... On page 7, line 174, we now state: ...about 200nm in width ($180\text{nm} \pm 50 \text{ nm}$, $n=97$),... On page 11, line 311, we now state ...the area that lacked fluorescence, between the two bands, was ~30% smaller in the mutant as compared to the wt (Suppl. Fig. S19).

Suppl. Fig. S19. Calculation of the area not covered by the PspC1 signal in the color map shown in Figure 7c. The distance between the PspC1 bands as the Full Width at half Minimum with respect to the lowest peak was calculated and was divided by the total length of the bacterium for BHN418 and the signal peptide mutant BHN418sp2pspC1. The length of the darker area as shown in the color map in Figure 7c was significantly larger in BHN418 in comparison to BHN418sp2pspC1.

We now hope that this reviewer find that we have made the statistic analyses and quantifications needed to support our conclusions.

Reviewer #3

The authors have done a great job to respond adequately to the reviewers' comments. They have performed additional experiments and now indicate by providing additional data the spatial distribution of the different PspC variants in the presence of the CPS. I have only two but minor comments:

We thank the reviewer for this positive comment stating that the authors have done a great job to respond adequately to the reviewers' comments.

1. the PspC models in the supplement (Fig. S3) can probably be modified in a way that not only the RICH are shown (which contain the FH binding motifs) but also binding region(s) of the polymeric Ig receptor. Studying the sequences the PspC1 contains two R domains and the RICH precedes these domains. The PspC2 variant contains two RICH domains – which seem to more or less directly after the signal peptide and N-terminal to the PRR, however, there is no R domain. This information should be included in the models and legends.

Response: We thank reviewer for this comments. We have now, as suggested by the reviewer, included the changes in Suppl. Fig. S4, see below.

Suppl. Fig. S4. Model structure of PspC1 and PspC2 from BHN418. PspC2 contains two RICH domains (Pfam-PF05062) in its N-terminal alpha helical part in comparison to one RICH domain in PspC1. R1 and R2 domains, containing conserved pIgR binding motif (represented by red lines), are present in PspC1, but absent in PspC2. Abbreviations: PRR-proline rich repeats, SP-signal peptide, CBD-choline binding domain, LPxTG-LPxTG anchor domain

2. the sensorgrams (not sensograms) of the Biacore data are raw data. The data have to be processed and the kinetic data should be shown.

Response: Because of the slow dissociation rate of PspC, we did single cycle kinetic (kinetic titrations) and the data presented are processed data (<https://www.sprpages.nl/data-fitting/models/kinetic-titration>) as described in the study by Karlsson et. al. (Karlsson, R., Katsamba, P.S., Nordin, H., Pol, E., and Myszka, D.G. Analyzing a kinetic titration series using affinity biosensors. Anal Biochem 2006;349, 136-147). To clarify this we have now added the home page and the reference in the Methods on page 17, line 508. We have also corrected the typo (Sensorgram) in Suppl. Fig. S6 as suggested by the reviewer.